# Biosensor for branched-chain amino acid metabolism in yeast and applications in isobutanol and isopentanol production

Yanfei Zhang[1], Jeremy D. Cortez[2,6], Sarah K. Hammer[1,6], César Carrasco-López [1], Sergio Á. García Echauri[1], Jessica B. Wiggins[3], Wei Wang [3] & José L. Avalos [1,2,4,5✉]

Branched-chain amino acid (BCAA) metabolism fulfills numerous physiological roles and can be harnessed to produce valuable chemicals. However, the lack of eukaryotic biosensors specific for BCAA-derived products has limited the ability to develop high-throughput screens for strain engineering and metabolic studies. Here, we harness the transcriptional regulator Leu3p from *Saccharomyces cerevisiae* to develop a genetically encoded biosensor for BCAA metabolism. In one configuration, we use the biosensor to monitor yeast production of isobutanol, an alcohol derived from valine degradation. Small modifications allow us to redeploy Leu3p in another biosensor configuration that monitors production of the leucine-derived alcohol, isopentanol. These biosensor configurations are effective at isolating high-producing strains and identifying enzymes with enhanced activity from screens for branched-chain higher alcohol (BCHA) biosynthesis in mitochondria as well as cytosol. Furthermore, this biosensor has the potential to assist in metabolic studies involving BCAA pathways, and offers a blueprint to develop biosensors for other products derived from BCAA metabolism.

---

[1] Department of Chemical and Biological Engineering, Princeton University, Princeton, NJ, USA. [2] Department of Molecular Biology, Princeton University, Princeton, NJ, USA. [3] Genomics Core Facility, Lewis-Sigler Institute for Integrative Genomics, Princeton University, Princeton, NJ, USA. [4] Andlinger Center for Energy and the Environment, Princeton University, Princeton, NJ, USA. [5] High Meadows Environmental Institute, Princeton University, Princeton, NJ, USA. [6] These authors contributed equally: Jeremy D. Cortez, Sarah K. Hammer. ✉email: javalos@princeton.edu

The metabolism of branched chain amino acids (BCAAs), including valine, leucine, and isoleucine is central to microbial production of many valuable products[1–8]. Among microbial hosts that produce BCAAs and BCAA-derived compounds, the yeast *Saccharomyces cerevisiae* is a favored industrial organism due to its genetic tractability, resistance to phage contamination, and ability to grow at low pH and high alcohol concentrations[9]. However, the rate at which genetic diversity can be introduced in strains greatly outpaces the rate at which they can be screened for production; and, unfortunately, there are currently no eukaryotic biosensors specific for individual BCAAs and their derived products, which could be used to develop high-throughput screens and accelerate strain development[10–15].

BCAA and BCAA-derived product biosynthesis originates from mitochondrial pyruvate[16] (Supplementary Fig. 1). In the biosynthesis of valine and leucine, an acetolactate synthase (ALS, encoded by *ILV2*) first catalyzes the condensation of two pyruvate molecules to produce acetolactate. The subsequent activities of a ketol-acid reductoisomerase (KARI, encoded by *ILV5*), and a dehydroxyacid dehydratase (DHAD, encoded by *ILV3*) yield α-ketoisovalerate (α-KIV), the precursor to valine. Next, α-KIV is acetylated in either mitochondria or the cytosol by α-isopropylmalate (α-IPM) synthases (encoded by *LEU4* and *LEU9*) to produce α-IPM. In the cytosol, α-IPM is converted by isopropylmalate isomerase (encoded by *LEU1*) and β-isopropylmalate dehydrogenase (encoded by *LEU2*) to α-ketoisocaproate (α-KIC), the precursor to leucine. Isoleucine biosynthesis resembles that of valine; however for isoleucine, Ilv2p condenses pyruvate with α-ketobutyrate derived from threonine. Subsequent Ilv5p and Ilv3p activity produces α-keto-3-methylvalerate (α-K3MV), the precursor to isoleucine. Finally, the three α-ketoacid precursors (α-KIV, α-KIC, and α-K3MV) are transaminated by branched-chain amino acid aminotransferases in mitochondria (encoded by *BAT1*) and cytosol (encoded by *BAT2*) to produce valine, leucine, and isoleucine, respectively[16] (Supplementary Fig. 1).

Many biosensors are based on transcription factors (TFs) that activate expression of a reporter gene, such as green fluorescent protein (GFP), in response to elevated concentrations of a specific molecule, which may be a product of interest, by-product, or precursor[12,13,17–19]. Leu3p is a yeast TF that regulates several genes in BCAA biosynthesis in response to selectively binding to α-IPM, acting as an activator in the presence of α-IPM and a repressor in its absence[20–22]. Because α-IPM is a by-product of valine biosynthesis and an intermediate of leucine biosynthesis (Supplementary Fig. 1), its levels are correlated to the activity of both pathways. This makes Leu3p an ideal TF to use in a biosensor for BCAA metabolic activity, which could enable high-throughput screens for production of BCAA-derived products such as branched-chain higher alcohols (BCHAs).

There is great interest in producing BCHAs, as they are among the top ten advanced biofuels identified by the U.S. Department of Energy for their potential to boost gasoline engine efficiency[23,24], and can be upgraded to renewable jet fuel[25]. BCHA production can be enhanced by overexpressing enzymes from the Ehrlich degradation pathway[26]. In this pathway, α-ketoacid decarboxylases (α-KDCs) and alcohol dehydrogenases (ADHs) convert the α-ketoacid precursors α-KIV, α-KIC, and α-K3MV to isobutanol, isopentanol, and 2-methyl-1-butanol, respectively (Supplementary Fig. 1).

Here, we report the development and application of a genetically encoded biosensor for eukaryotic BCAA metabolism and BCHA biosynthesis. The biosensor is based on the α-IPM-dependent function of Leu3p to control expression of GFP. Two biosensor configurations enable high-throughput screening for improved isobutanol or isopentanol production. These two configurations differ mechanistically, by monitoring α-IPM as either a by-product of isobutanol or a precursor of isopentanol. We apply the biosensor to isolate high-producing strains, identify mutant enzymes with enhanced activity, and construct biosynthetic pathways for production of isobutanol and isopentanol in both mitochondria and cytosol. This biosensor has the potential to accelerate the development of yeast strains to produce BCHAs as well as other products derived from BCAA metabolism.

## Results

**Design and construction of a biosensor for branched-chain higher alcohol biosynthesis.** To construct a biosensor for BCHAs, isobutanol and isopentanol, we introduced a copy of yeast-enhanced green fluorescent protein (yEGFP) under the control of the Leu3p-regulated *LEU1* promoter ($P_{LEU1}$) and a constitutively expressed leucine-insensitive Leu4p mutant. We tested 8 different biosensor constructs using 4 different Leu4p variants, and the yEGFP tagged or untagged with a proline-glu-tamate-serine-threonine-rich (PEST) protein degradation tag[27] to test different stability levels of the reporter. We compared the change in fluorescence caused by overexpressing the BCHA biosynthetic pathway in strains with or without a *LEU2* deletion, which respectively favors isobutanol or isopentanol production, in a total of 16 different strains (Supplementary Note 1). When *LEU2* is deleted, the construct using a Leu4p variant with a truncated leucine regulatory domain (Leu4$^{1–410}$) and a PEST-tagged yEGFP gives the largest change in fluorescence (Supplementary Fig. 2a). The better performance of the PEST-tagged yEGFP is consistent with the expected accumulation of IPM molecules when *LEU2* is deleted (Supplementary Fig. 1), which would boost Leu3p activation and lead to higher background (Supplementary Note 1). In contrast, when *LEU2* is present, having a Leu4p variant with a Ser547 deletion (Leu4$^{ΔS547}$)[28] and untagged yEGFP shows the largest difference in fluorescence (Supplementary Fig. 2b). This is consistent with lower α-IPM concentrations due to Leu2p activity, which lowers Leu3p background activation and thus requires higher stability of yEGFP for enhanced biosensor sensitivity. We also found that deleting the endogenous *LEU4* and *LEU9* reduces the biosensor background in *LEU2* strains (Supplementary Note 1). Because *LEU2* favors isopentanol production, while *LEU2* deletion favors isobutanol, the constructs containing yEGFP-PEST/*LEU4*$^{1–410}$/*leu2Δ* (henceforth called the isobutanol configuration) and yEGFP/Leu4$^{ΔS547}$/*LEU2*/*leu4Δ*/ *leu9Δ* (henceforth called the isopentanol configuration) make suitable biosensor configurations to screen for improved isobutanol and isopentanol production, respectively (Fig. 1a, c).

To characterize the biosensor in the isobutanol configuration, we measured its response to key metabolites supplemented in the growth medium as well as to enhanced isobutanol biosynthesis in engineered strains. We found that the fluorescence emitted by the biosensor responds linearly to increasing concentrations of α-IPM ($R^2 = 0.96$, Supplementary Fig. 3a) or α-KIV ($R^2 = 0.97$, Supplementary Fig. 3b) supplemented in the media at concentrations below 80 μM. At higher concentrations, the response loses linearity, possibly due to cellular import bottlenecks of α-IPM and α-KIV (Supplementary Fig. 3c, d, and Supplementary Note 2). In contrast, the biosensor does not respond to the isoleucine precursor α-K3MV, when supplemented in the media (Supplementary Fig. 3e, f). We next examined the correlation between GFP fluorescence and isobutanol production in strains containing the biosensor and different constructions of the BCHA biosynthetic pathway, resulting in varying levels of isobutanol production. We found that there is a strong linear correlation ($R^2 = 0.97$) between GFP fluorescence during the exponential phase and total isobutanol biosynthesis (Fig. 1b), providing a

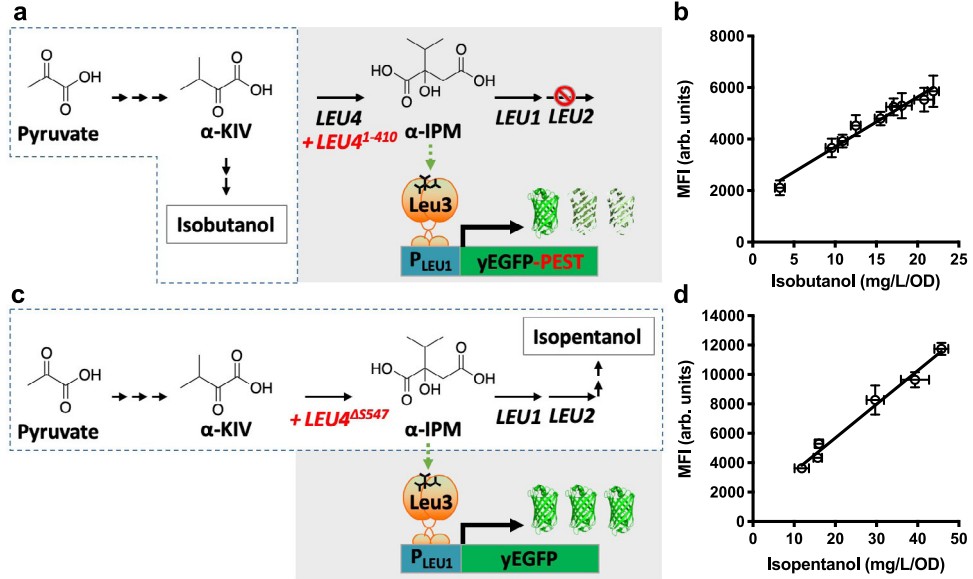

**Fig. 1 Design and characterization of two BCHA biosensor configurations. a** Schematics of the biosensor in the isobutanol configuration (gray box) and simplified isobutanol pathway (enclosed by dashed lines; each arrow represents one enzymatic step). The α-isopropylmalate (α-IPM), which activates Leu3p (green dashed arrow), is an isobutanol by-product that accumulates due to deletion of *LEU2*. Thus, adding a PEST tag to destabilize yEGFP reduces background. α-KIV, α-ketoisovalerate. **b** Correlation between specific isobutanol titers (mg/L/OD) produced by engineered strains and GFP median fluorescence intensity (MFI) from the isobutanol biosensor. Strains used (from left to right): YZy121, YZy230, YZy81, YZy231, YZy232, YZy233, YZy234, YZy235, and YZy236 (Supplementary Table 1). **c** Schematics of the biosensor in the isopentanol configuration (gray box) and simplified isopentanol pathway (enclosed by dashed lines; each arrow represents one enzymatic step). In this case, α-IPM is an isopentanol precursor that does not accumulate due to *LEU2* activity, so the PEST tag on yEGFP is not required. **d** Correlation between specific isopentanol titers (mg/L/OD) produced by engineered strains and GFP median fluorescence intensity (MFI) from the isopentanol biosensor. Strains used (from left to right): SHy187, SHy188, SHy192, SHy159, SHy176, and SHy158 (Supplementary Table 1). The MFI for each strain was measured 13 h after inoculation, during the exponential phase, and plotted against the corresponding specific isobutanol or isopentanol titers obtained after 48-h high-cell-density fermentations. MFI are represented in arbitrary units (arb. units). All data are shown as mean values. Error bars represent the standard deviation of at least three biological replicates. Source data are provided as a Source Data file.

conservative estimate of its dynamic range (Supplementary Note 2). Moreover, measuring the intracellular α-IPM concentrations of an isobutanol-producing strain against a control strain (see Methods) revealed that Leu3p-sensing of α-IPM concentrations during exponential growth is predictive of the total isobutanol produced by the end of the fermentation (Supplementary Fig. 4a–c and Supplementary Note 3). These results validate the applicability of this biosensor to measure isobutanol production in yeast.

We next validated the efficacy of the biosensor in the isopentanol configuration. To achieve this, we measured the fluorescence of several *LEU2* strains containing this biosensor configuration and various constructs of the BCHA biosynthetic pathway, resulting in different levels of isopentanol production. We found a strong linear correlation ($R^2 = 0.98$) between the biosensor fluorescence during the exponential phase and total isopentanol production by the end of the fermentation (Fig. 1d and Supplementary Note 2). Time-course measurements of fluorescence and intracellular α-IPM concentrations of an isopentanol-producing strain revealed that the biosensor in this configuration is highly sensitive to small differences in α-IPM (Supplementary Fig. 4d–f). Interestingly, fluorescence measurements of strains with the biosensor in the isopentanol configuration do not correlate to isobutanol production (Supplementary Fig. 5). Therefore, mechanistic differences between the biosensor configurations allow them to measure specifically either isobutanol or isopentanol production (Supplementary Note 3).

**Using the biosensor to isolate high-isobutanol-producing strains**. We applied the isobutanol configuration of the

biosensor to isolate the highest isobutanol producers from a library of strains containing random combinations of genes from the mitochondrial isobutanol biosynthesis pathway[29]. Equal molar amounts of cassettes containing either upstream *ILV* genes (*ILV2, ILV3*, and *ILV5*), downstream Ehrlich pathway genes (KDC and ADH), or the full isobutanol pathway, all designed to randomly integrate into YARCdelta5 δ-sites[29,30], were pooled and used to transform strain YZy81, which has the biosensor in the isobutanol configuration (Supplementary Tables 1, 2). After subjecting the transformed population to two rounds of fluorescence activated cell sorting (FACS, see Methods), 11 out of 24 randomly chosen colonies produce more than 600 mg/L isobutanol. This is more than twice the mean isobutanol titer (268 mg/L) obtained from 24 random colonies from the unsorted population; in fact, none of the unsorted colonies produced more than 600 mg/L and only one produced more than 500 mg/L (Fig. 2). Additionally, using digital droplet PCR (see methods), we found only two different genotypes in the top 6 isobutanol-producing strains (those producing 700 mg/L isobutanol or more), suggesting a substantial enrichment from the ~$2 \times 10^5$ individual transformants in the original library (Supplementary Table 3). These results demonstrate that the isobutanol biosensor configuration enables high-throughput screening with FACS to isolate high-isobutanol producers from mixed strain populations.

**Applying the isobutanol configuration of the biosensor to identify ILV6 mutants insensitive to valine inhibition**. We next utilized the isobutanol biosensor configuration to identify mutants of the acetolactate synthase regulatory protein, encoded by *ILV6*, with reduced feedback inhibition from valine. Ilv6p

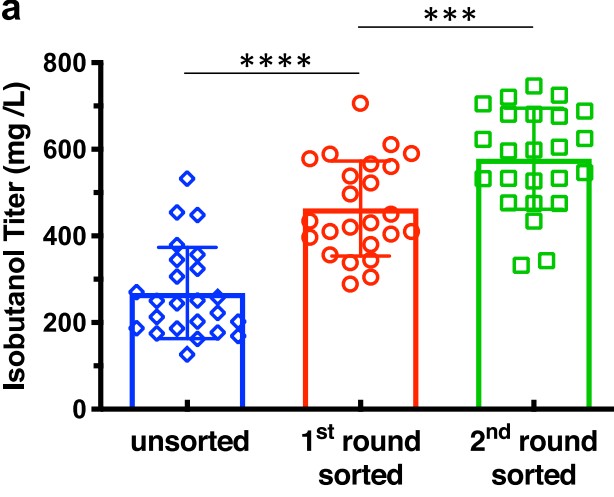

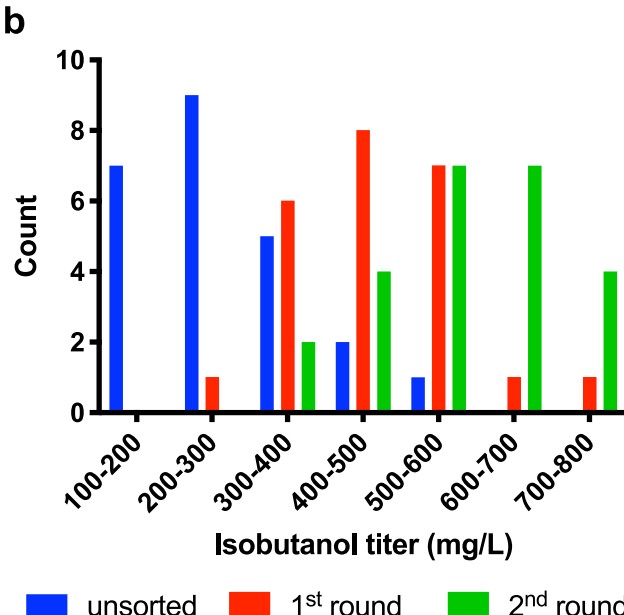

**Fig. 2 Applying the biosensor in its isobutanol configuration for high-throughput screening of a library of strains combinatorially transformed with the mitochondrial isobutanol biosynthetic pathway. a** Isobutanol titers of 24 colonies randomly selected from unsorted (blue), first round sorted (red), or second round sorted (green) populations derived from a library of strains transformed with constructs overexpressing different combinations of the mitochondrial isobutanol pathway. All data are shown as mean values. Error bars represent the standard deviation of the titers of the 24 colonies analyzed. A two-sided t-test was used to determine the statistical significance of the differences in isobutanol titers of analyzed strains from each population; From left to right: $P < 0.0001$, $P = 0.001$; ***$P \leq 0.001$, ****$P \leq 0.0001$. **b** Histogram showing the number of colonies exhibiting different ranges of isobutanol production, out of the same 24 randomly selected strains from the unsorted (blue), first round sorted (red), or second round sorted (green) populations shown in (**a**). Source data are provided as a Source Data file.

localizes to mitochondria where it interacts with acetolactate synthase, Ilv2p, both enhancing its activity and bestowing upon it feedback inhibition by valine[31,32] (Fig. 3a). Mutations identified in the Ilv6p homolog from *Streptomyces cinnamonensis* (IlvN*)* that confer resistance to valine analogues[33] informed the design of an Ilv6p[V90D/L91F] mutant, which has been shown to retain

Ilv2p activation, have reduced sensitivity to valine inhibition, and improve isobutanol production[34,35]. However, these mutations are likely not unique or optimal. If alternative mutations exist, some potentially conferring higher Ilv2p/Ilv6p complex activity, we hypothesized that we could find them with the aid of our BCHA biosensor.

To find Ilv6p mutants insensitive to valine inhibition, we used the isobutanol biosensor configuration to isolate highly fluorescent strains transformed with libraries of randomly mutagenized *ILV6* and grown in the presence of valine. To avoid biosensor signal interference, the parent strain of these libraries (YZy91) contains *BAT1* and *BAT2* deletions, which eliminates interconversion between valine and α-KIV, as well as an *ILV6* deletion to rule out endogenous valine inhibition of Ilv2p (Supplementary Table 1, Fig. 3a). Two libraries, generated via error-prone PCR of either the wild-type *ILV6* or the *ILV6[V90D/L91F]* variant, were used separately to transform YZy91. After culturing the two resulting yeast libraries in media containing excess valine (see Methods) and three rounds of FACS, we measured the isobutanol production of 24 colonies randomly picked from each sorted library. All 48 colonies randomly selected after the third round of sorting show higher fluorescence and isobutanol production in the presence of valine than the strain overexpressing wild-type *ILV6* (Supplementary Fig. 6), demonstrating an undetectable low rate of false positives.

We next analyzed the *ILV6* mutations found in the sorted strains. First, we tested whether each *ILV6* mutant was responsible for the enhanced isobutanol production and biosensor signal by curing each strain of the plasmid that contains it (expecting a return to parental phenotypes), as well as by re-transforming the parental strain (YZy91) with each isolated plasmid (expecting a return to enhanced phenotypes). These experiments confirmed that the *ILV6* mutants indeed confer the enhanced phenotypes (Fig. 3b and Supplementary Fig. 7), although the retransformation experiments in some strains suggest that additional background mutations attenuate (e.g., strain 8) or boost (e.g., strain 11) the effects of these mutants. Sequencing the 48 isolated plasmids revealed 24 unique sequences and several previously unknown *ILV6* mutations that achieve similar or higher levels of isobutanol production than the previously known *ILV6[V90D/L91F]* (Supplementary Table 4). Variant 6, with a single mutation at Val110 (V110E), produces the most isobutanol (378 mg/L ± 10 mg/L), which is a 3.2-fold increase over the wild-type *ILV6*, and a small but statistically significant ($P = 0.0055$) improvement over the previously identified *ILV6[V90D/L91F]* mutant (Fig. 3b). When we overexpress this *ILV6[V110E]* variant in a strain containing the mitochondrial isobutanol biosynthetic pathway, the production is 1.6 times higher than the equivalent strain overexpressing the wild-type *ILV6* (Fig. 3c). Mapping the mutations found in our isolated variants onto the crystal structure of the bacterial Ilv6p homologue IlvH revealed that most of the mutations are located in the regulatory valine binding domain (Supplementary Note 4 and Supplementary Fig. 8). Although the biosensor is able to aid in identifying both previously reported and unknown mutations that increase isobutanol production relative to wild-type *ILV6*, none of the previously unknown mutants achieve a substantial improvement in isobutanol production relative to *ILV6[V90D/L91F]*. This could be because *ILV6[V90D/L91F]* enhances Ilv2p activity to an extent that is near the maximum achievable through Ilv6p engineering alone, or because the pathway bottleneck is no longer at the Ilv2p metabolic step after *ILV6[V90D/L91F]* is overexpressed.

**Applying the isopentanol configuration of the biosensor to identify highly active LEU4 mutants.** We next applied our

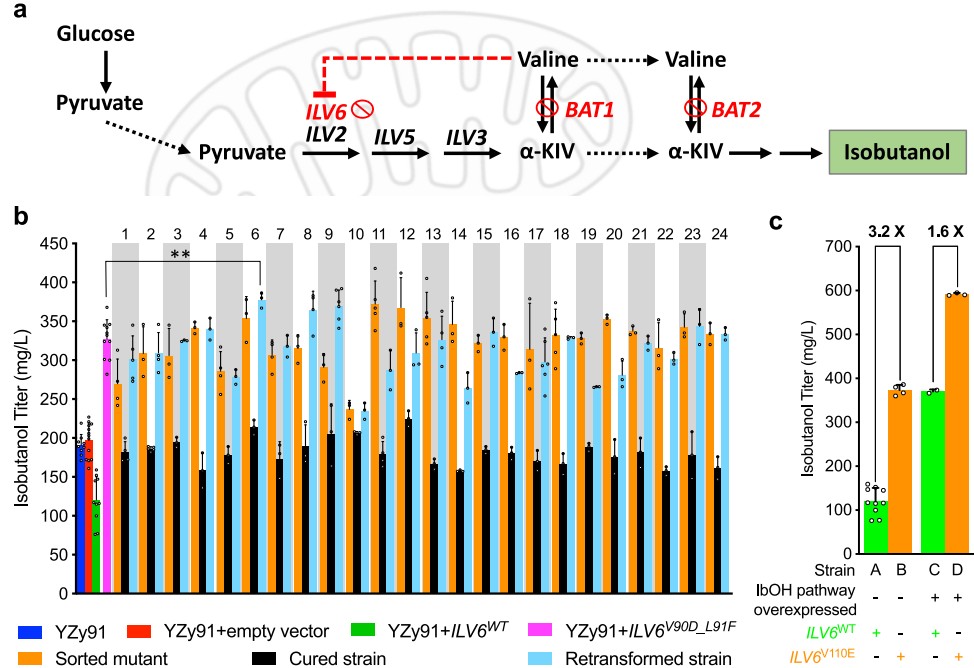

**Fig. 3 High-throughput screen for metabolically hyperactive Ilv6p mutants using the biosensor in its isobutanol configuration. a** Schematic representation of the genotype of the basal strain, YZy91, used for high-throughput screening of Ilv6p mutants with enhanced activity in the presence of valine; α-KIV: α-ketoisovalerate. Mitochondrion image created with BioRender. **b** Isobutanol titers after 48 h fermentations of sorted strains with unique *ILV6* mutant sequences (orange), plasmid-cured derivatives (black), and strains obtained from retransforming YZy91 with each unique plasmid isolated from the corresponding sorted strain (cyan). Titers obtained from YZy91 with (red) or without (blue) an empty vector, or transformed with plasmids containing *ILV6*$^{WT}$ (green) or *ILV6*$^{V90D\_L91F}$ (magenta), are shown as controls. The numbers on the upper x-axis identify each of the strains harboring screened mutants with unique *ILV6* sequences. *ILV6* mutants 1–10 were isolated from the sorted wild-type *ILV6* (*ILV6*$^{WT}$) mutagenesis library. *ILV6* mutants 11–24 were isolated from the sorted *ILV6*$^{V90D\_L91F}$ mutagenesis library. A two-sided *t*-test was used to determine the statistical significance of the difference in isobutanol titers between YZy91 with *ILV6*$^{V90D\_L91F}$ (magenta) and *ILV6*$^{V110E}$ (*ILV6* mutant #6). P = 0.0055; **P ≤ 0.01. **c** Isobutanol titers of YZy91 transformed with CEN plasmids containing *ILV6*$^{WT}$ (Strain A) or *ILV6*$^{V110E}$ (*ILV6* mutant #6, Strain B) compared to titers from a strain overexpressing the mitochondrial isobutanol pathway (YZy363) transformed with CEN plasmids containing *ILV6*$^{WT}$ (Strain C) or *ILV6*$^{V110E}$ (*ILV6* mutant #6, Strain D). All data are shown as mean values. Open circles represent individual data points. Error bars represent the standard deviation of at least three biological replicates. Source data are provided as a Source Data file.

biosensor in the isopentanol configuration to identify *LEU4* mutants with enhanced enzymatic activity. While the isopentanol configuration utilizes a *LEU4* mutant previously shown to have reduced sensitivity to leucine inhibition[28,36], we sought to identify a variant that is not only insensitive to leucine inhibition, but more catalytically active. To do so, we constructed two overexpression libraries of randomly mutagenized *LEU4* (using error-prone PCR): one with full-length *LEU4* mutagenized, and another with only the leucine regulatory domain mutagenized (residues 430–619). These libraries were separately used to transform a ΔBAT1/ΔLEU4/ΔLEU9 strain containing a modified isopentanol configuration of the biosensor lacking the *LEU4*$^{ΔS547}$ (YZy148, Supplementary Tables 1 and 2). Two major sub-populations in the full-length *LEU4* library (ep_Leu4) were observed during growth in the presence of leucine (Fig. 4a). The larger sub-population overlaps with the signal from the empty vector control, suggesting that most mutations are deleterious to Leu4p activity, and thus α-IPM synthesis. However, the smaller sub-population retains activity, with a substantial fraction exhibiting higher GFP fluorescence intensity than the wild-type *LEU4*. In contrast, the library derived from mutagenizing only the regulatory domain of *LEU4* (ep_Leu4_Reg) is dominated by a single population with a higher median GFP signal than wild-type *LEU4* (Fig. 4a), suggesting that many mutations in the regulatory domain enhance Leu4p enzymatic activity in the presence of leucine.

After three rounds of FACS, both sorted populations display increased median fluorescence intensity (Fig. 4b). The isopentanol production of 48 colonies randomly selected from the third

round of sorting ranged between 375 mg/L and 725 mg/L (Fig. 4c and Supplementary Fig. 9), which represent 1.9- to 3.6-fold improvements over the strain overexpressing the wild-type *LEU4*. Similar to the isobutanol configuration, these results indicate that the isopentanol configuration displays a low (undetectable) rate of false positives. Taking the same approach as in the *ILV6* experiments above, we cured each strain from its *LEU4*-containing plasmid and retransformed the parent strain (YZy148) with the same extracted plasmids to compare fluorescence intensity (Supplementary Fig. 10) and isopentanol production (Fig. 4c) between the sorted, cured, and retransformed strains. Notably, the strain retransformed with mutant #6 achieves the highest isopentanol titer (963 mg/L ± 32 mg/L), which is 4.8-times higher than the titer achieved by overexpressing the wild-type *LEU4*, and significantly higher than the titer obtained with the *LEU4*$^{ΔS547}$ mutant previously described[37] (842 mg/L ± 23 mg/L; P-value = 0.0021).

Sequencing the plasmids contained in the 48 randomly selected colonies (Supplementary Fig. 9), revealed 24 unique *LEU4* variants (Supplementary Table 5). Several newly found mutations may contribute to reduced regulation, enhanced catalytic activity, or both. Five variants have single mutations (H541R, Y485N, Y538N, V584E, and T590I), all of which are located in the regulatory domain of Leu4p (Supplementary Note 5 and Supplementary Fig. 11). Although we have not yet elucidated the combined effects of multiple amino acid substitutions in variants containing two or more mutations, seven residues (K51,

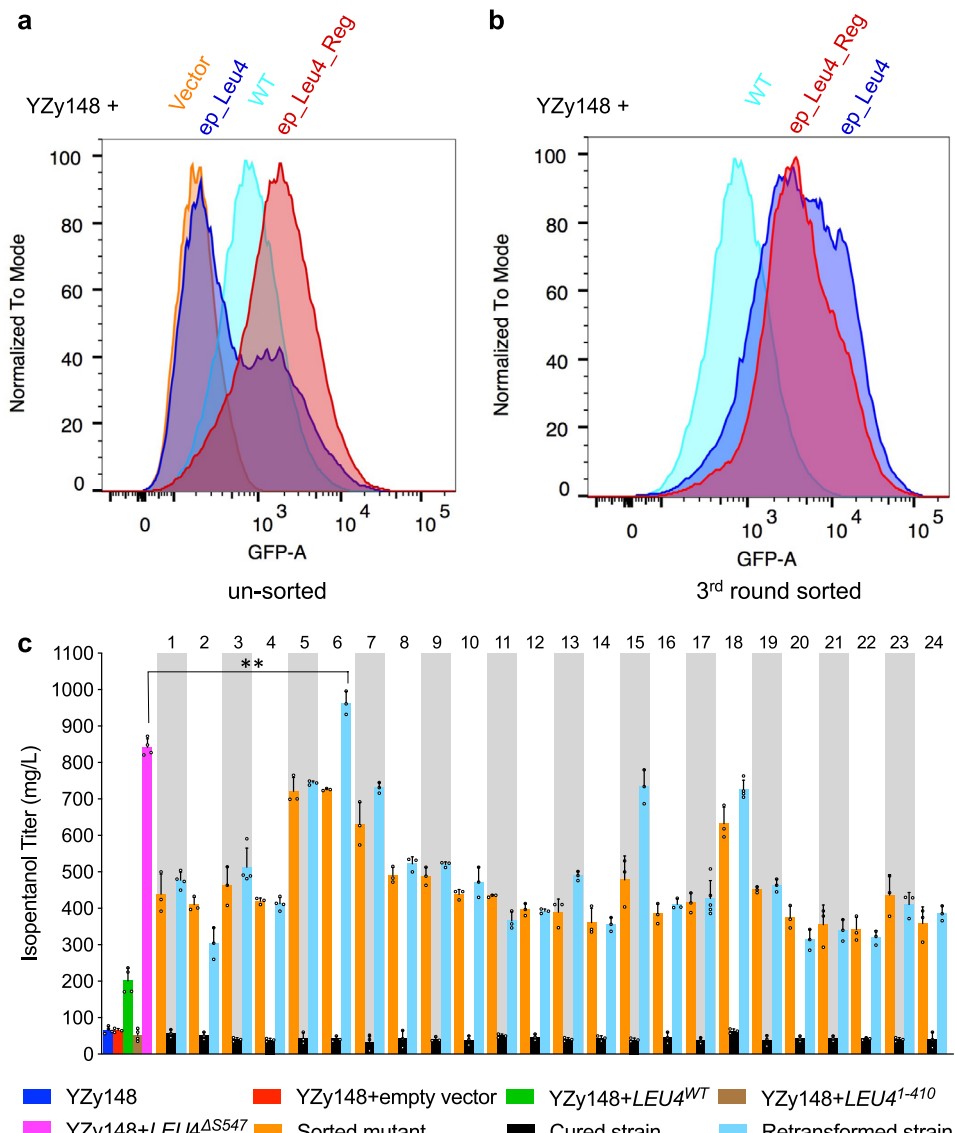

**Fig. 4 High-throughput screen for hyperactive Leu4p variants using the isopentanol configuration of the biosensor. a** Flow cytometry analysis of yeast libraries containing *LEU4* variants generated with error-prone PCR-based mutagenesis of the full-length *LEU4* (ep_Leu4, blue) or only the *LEU4* regulatory domain (ep_Leu4_Reg, red), compared to control strains containing an empty vector (Vector, orange) or wild-type *LEU4* (WT, cyan). Cell populations were normalized to the mode for comparison. **b** Flow cytometry analysis of the same libraries depicted in (**a**) after three rounds of FACS. **c** Isopentanol titers after 48 h fermentations of sorted strains with unique *LEU4* sequences (orange), plasmid-cured derivatives (black), and strains obtained from retransforming YZy148 with each unique plasmid isolated from the corresponding sorted strain (cyan). Titers obtained from the basal strain YZy148 (*leu4Δ leu9Δ bat1Δ LEU2*) with (red) or without (blue) an empty vector, or transformed with plasmids containing wild-type *LEU4* (*LEU4^WT*, green), *LEU4^1–410* (brown), or *LEU4^ΔS547* (magenta), are shown as controls. The numbers on the upper x-axis identify each strain harboring a unique *LEU4* sequence. *LEU4* mutants 1–13 are derived from mutagenizing the full-length *LEU4*; and mutants 14–24 from mutagenizing the *LEU4* leucine regulatory domain. All data are shown as mean values. Open circles represent individual data points. Error bars represent the standard deviation of at least three biological replicates. A two-sided *t*-test was used to determine the statistical significance of the difference between isopentanol titers achieved by YZy148 transformed with a plasmid containing *LEU4^ΔS547* (magenta), or *LEU4* mutant #6 (*LEU4^E86D_K191N_K374R_A445T_S481R_N515I_A568V_S601A*). $P = 0.0021$; $**P \leq 0.01$. Source data are provided as a Source Data file.

Q439, F497, N515, V584, D578, and T590) are substituted in more than one variant (Supplementary Tables 5, 6), suggesting they are involved in enhancing Leu4p activity. Thus, our biosensor was able to identify 12 previously unreported residues in Leu4p that enhance isopentanol production when mutated (Supplementary Note 5).

**Biosensor-assisted cytosolic isobutanol pathway engineering.** The demonstrations above involve strains engineered with the

BCHA biosynthetic pathway compartmentalized in mitochondria[29]; however, cytosolic pathways for isobutanol production have also been developed[38–41]. To investigate whether the isobutanol-configured biosensor can help improve pathways compartmentalized in either compartment, we used it to optimize an isobutanol pathway contained entirely in the cytosol, comprised of heterologous ALS, KARI, and DHAD (*ILV*) genes. We deleted *ILV3* from strain YZy121 (Supplementary Table 1) to eliminate mitochondrial α-KIV synthesis and ensure the biosensor signal and isobutanol production are derived only from cytosolic activity (Fig. 5a). We

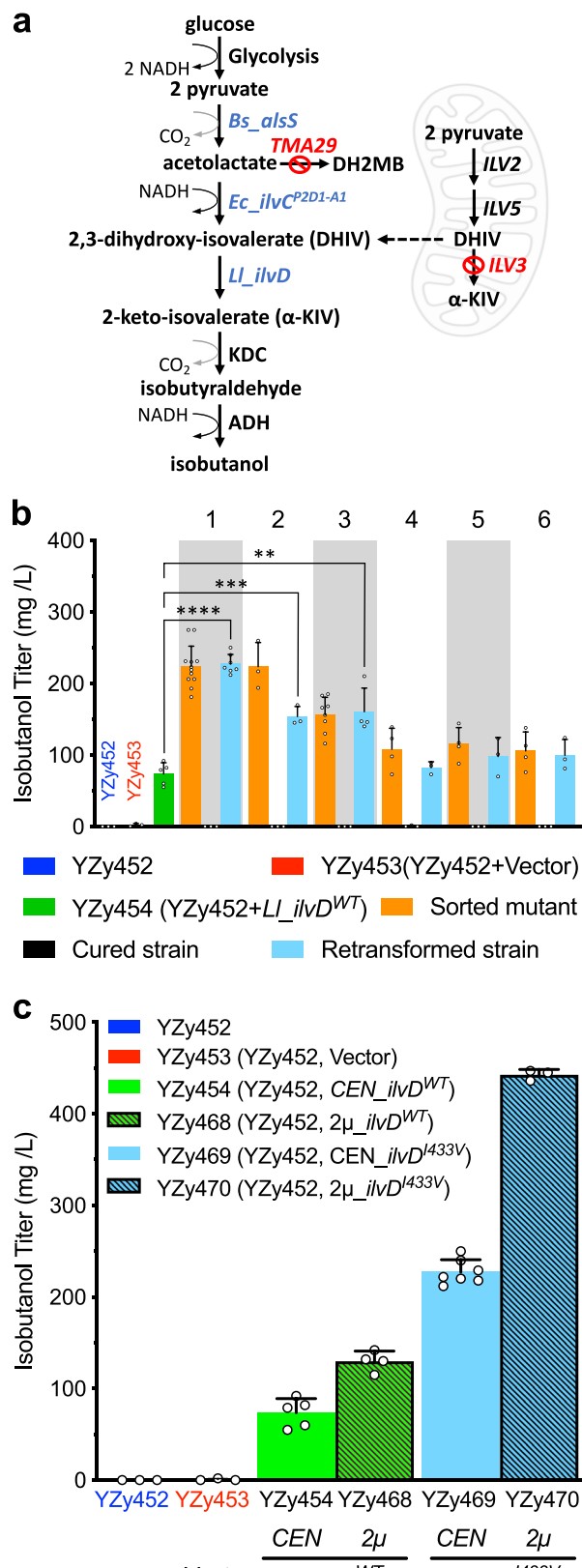

**Fig. 5 Biosensor-assisted cytosolic isobutanol pathway engineering.**
**a** Schematic of the engineered cytosolic isobutanol pathway. *Bs_alsS*: acetolactate synthase (ALS) from *Bacillus subtilis*; *Ec_ilvC*$^{P2D1-A1}$: NADH-dependent ketol-acid reductoisomerase (KARI) variant from *E. coli*; *Ll_ilvD*: dihydroxyacid dehydratase (DHAD) from *Lactococcus lactis*; DH2MB: 2,3-dihydroxy-2-methyl butanoate. Mitochondrion image created with BioRender. **b** Isobutanol titers after 48 h fermentations in 15% glucose of sorted strains with unique *Ll_ilvD* sequences (orange), plasmid-cured derivatives (black), and strains obtained from retransforming YZy452 with each unique plasmid isolated from the corresponding sorted strain (cyan). The numbers on the upper *x*-axis identify each of the strains harboring screened mutants with unique *Ll_ilvD* sequences. Titers from YZy452 (containing the cytosolic isobutanol pathway with extra copies of *Ec_ilvC*$^{P2D1-A1}$) with (red) or without (blue) an empty vector, or transformed with a plasmid containing wild-type *Ll_ilvD* (green) are shown as controls. A two-sided *t*-test was used to determine the statistical significance of the difference between isobutanol titers achieved by YZy452 transformed with a plasmid containing *Ll_ilvD*$^{WT}$ (green), or *Ll_ilvD* mutant #1 (*Ll_ilvD*$^{I433V}$), or mutant #2 (*Ll_ilvD*$^{V12A, S189P, H439R}$), or mutant #3 (*Ll_ilvD*$^{K535R}$). From left to right: $P < 0.0001$, $P = 0.0003$, $P = 0.0011$; **$P \le 0.01$, ***$P \le 0.001$, ****$P \le 0.0001$. **c** Isobutanol titers of the basal strain YZy452 (blue) transformed with an empty vector (red), *Ll_ilvD*$^{WT}$ (green), or *Ll_ilvD*$^{I433V}$ (cyan), using CEN (solid) or 2μ (striped) plasmids. Isobutanol production was measured after 48 h fermentations in 15% glucose. All data are shown as mean values. Open circles represent individual data points. Error bars represent the standard deviation of at least three biological replicates. Source data are provided as a Source Data file.

encoding a DHAD from *Lactococcus lactis* (Fig. 5a). Because *Ll*_IlvD requires iron-sulfur (Fe/S) cluster cofactors, we also over-expressed *AFT1*, a transcription factor involved in iron utilization and homeostasis[45]. The resulting strain (YZy449) contained a metabolic pathway to convert pyruvate to α-KIV in the cytosol, which can then be converted to isobutanol by endogenous cytosolic KDCs and ADHs (Fig. 5a).

We applied the isobutanol biosensor to find strains with increased expression of *Ec_ilvC*$^{P2D1-A1}$ that enhance cytosolic isobutanol production. Given that the catalytic rate constant ($k_{cat}$) of *Bs*_AlsS is about 28 times higher than that of *Ec_ilvC*$^{P2D1-A1}$ (Supplementary Table 7), we hypothesized that overexpressing additional copies of *Ec_ilvC*$^{P2D1-A1}$ would compensate for this difference and improve isobutanol production. Thus, we randomly integrated a cassette containing two copies of *Ec_ilvC*$^{P2D1-A1}$ (pYZ206) into YARCdelta5 δ-sites of YZy449 (Supplementary Tables 1, 2), and sorted for transformants with the highest fluorescence intensity. After three rounds of FACS, 20 randomly picked colonies produce an average of 296 ± 19 mg/L isobutanol from 15% galactose, with the highest producing strain (YZy452) achieving 310 ± 15 mg/L isobutanol (Supplementary Fig. 12). In contrast, 20 randomly picked colonies from the unsorted population produce on average only 188 ± 19 mg/L (Supplementary Fig. 12). These results demonstrate that the isobutanol-configured biosensor can identify strains with increased cytosolic isobutanol production.

We next explored if the isobutanol-configured biosensor could help identify *Ll_ilvD* variants that further enhance isobutanol production. Beginning with the highest producing strain isolated as described above (YZy452), we introduced an error-prone PCR library of *Ll_ilvD* mutants in CEN plasmids. Because the wild-type copy of *Ll_ilvD* in YZy452 is expressed from a P$_{GAL10}$ promoter, growing the library in glucose ensures this wild-type copy is repressed, and only the *Ll_ilvD* variants from the CEN plasmid library (inserted downstream of a P$_{TDH3}$ promoter) are expressed. After three rounds of FACS, all 24 randomly picked

also deleted *TMA29*, encoding an enzyme with acetolactate reductase activity that competes with KARI, producing the by-product 2,3-dihydroxy-2-methyl butanoate (DH2MB)[42]. We then integrated in the resulting strain (YZy443) a single copy of *Bs_alsS*, encoding an ALS from *Bacillus subtilis*[43], *Ec_ilvC*$^{P2D1-A1}$, encoding an engineered NADH-dependent KARI from *E. coli*[44], and *Ll_ilvD*,

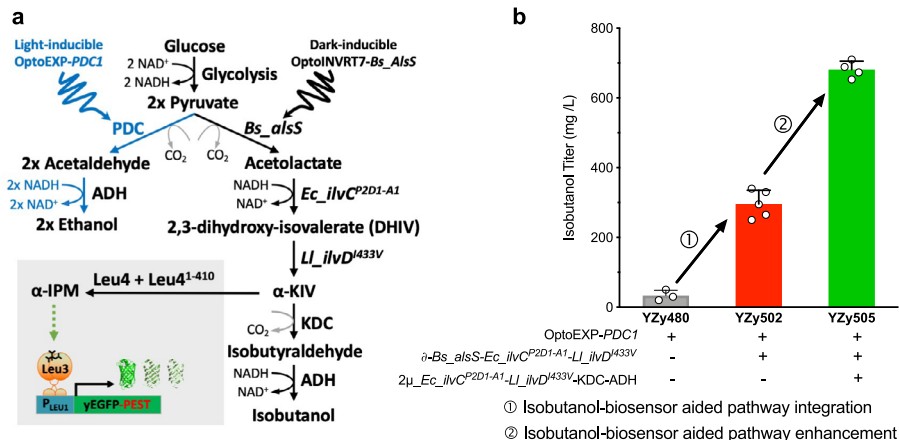

**Fig. 6 Biosensor-assisted engineering of optogenetically controlled strains for cytosolic isobutanol production. a** Schematic of the dark-inducible cytosolic isobutanol pathway (with Bs_alsS controlled by OptoINVRT7) and light-inducible PDC1 (controlled by OptoEXP) in a triple pdcΔ strain containing the isobutanol configuration of the biosensor. **b** Isobutanol production of optogenetically regulated strains, including a basal strain containing only light-inducible PDC1 (YZy480, gray); a strain also containing the dark-inducible upstream cytosolic pathway (Bs_alsS, Ec_ilvC^P2D1-A1, and Ll_ilvD^I433V) integrated in δ-sites (YZy502, red); and a strain containing additional enzymes from the cytosolic isobutanol pathway (Ec_ilvC^P2D1-A1, Ll_ilvD^I433V, KDC, and ADH) introduced in a 2μ plasmid (YZy505, green). Isobutanol production was measured after 48 h fermentations in media containing 2% glucose. Strains YZy502 and YZy505 where each isolated after two rounds of FACS. All data are shown as mean values. Open circles represent individual data points. Error bars represent the standard deviation of at least three biological replicates. Source data are provided as a Source Data file.

colonies produce between 150 mg/L and 260 mg/L isobutanol from 15% glucose, which is 2- to 3.5-times higher than a strain (YZy454) containing a plasmid with the wild-type Ll_ilvD (Supplementary Fig. 13a). Among the 24 random colonies examined, we found six unique Ll_ilvD sequences, most of them containing the sequence of mutant #1 (I433V, Supplementary Table 8). As described above for Ilv6p and Leu4p variants, fluorescence intensity measurements (Supplementary Fig. 13b) and isobutanol production (Fig. 5b) of sorted, cured, and retransformed strains were compared. Four of the six retransformed strains reproduce the isobutanol titers of the sorted strains, confirming the Ll_ilvD variants they contain cause the enhanced phenotypes and suggesting that background mutations in the two remaining sorted strains are responsible for them. The strain retransformed with the most active variant Ll_ilvD^I433V (YZy469) produces 227 ± 13 mg/L of isobutanol from 15% glucose, which is 3.1-fold higher than the strain harboring the wild-type Ll_ilvD (YZy454, Fig. 5c). When Ll_ilvD^I433V is overexpressed using a 2μ plasmid (YZy470), the titer further increases to 443 mg/L ± 6 mg/L, which is 3.4-fold higher than overexpressing the wild-type Ll_ilvD in a 2 μ plasmid (YZy468, Fig. 5c).

To gain structural insights on possible mechanisms by which Ll_ilvD activity is enhanced, we mapped the mutations in the most active variants onto the crystal structure of an IlvD homolog (Supplementary Fig. 14). Ll_IlvD mutants #1 and #3 each have a single mutation, I433V and K535R, respectively; however, mutant #2 has three residue substitutions (V12A, S189P, and H439R), which makes it more difficult to ascertain the importance and contribution of each mutation in this variant. Interestingly, all five mutations in the three variants are solvent exposed (Supplementary Fig. 14a, b), suggesting that some of them may improve the solubility, expression, or stability of the enzyme. In addition, mutations I433V and S189P, are located on opposing lobes that define the putative substrate entrance to the active site (Supplementary Fig. 14c, d), suggesting these mutations may enhance enzymatic activity by affecting the dynamics of opening and closing of the active site (Supplementary Note 6). In summary, our biosensor is capable of identifying previously

unknown enzyme mutations that not only enhance isobutanol production, but may also improve our understanding of the molecular mechanisms that limit enzyme activity.

Finally, we set out to show that our biosensor could work in conjunction with optogenetic actuators to select for strains with enhanced flux through cytosolic isobutanol synthesis. We previously showed that optogenetically controlling the expression of the pyruvate decarboxylase (PDC1) and the mitochondrial isobutanol pathway in a triple PDC deletion background (pdc1Δ pdc5Δ pdc6Δ) boosts isobutanol production[46]. This is an effective way to increase metabolic flux towards pathways of interest that compete with PDC1 without completely eliminating pyruvate decarboxylase activity, which causes a severe growth defect in glucose[47,48]. We tested our biosensor in a strain in which PDC1 is controlled by OptoEXP[46], an optogenetic circuit that induces expression only in blue light (450 nm), and the first enzyme of the cytosolic pathway (Bs_AlsS) is controlled by OptoINVRT7[49], an optogenetic circuit that represses gene expression in blue light and activates it in the dark (Fig. 6a). The starting strain for this experiment (YZy502) has a triple PDC deletion background with a light-inducible PDC1, a dark-inducible cytosolic isobutanol pathway, and our biosensor in its isobutanol configuration (Supplementary Table 1). We transformed YZy502 with a 2μ plasmid (pYZ350, Supplementary Table 2) containing two copies of Ec_ilvC^P2D1-A1 and one copy each of Ll_IlvD^I433V, ARO10 (KDC), and Ll_adhA^RE1(ADH)[50]. Using the biosensor in combination with optogenetic controls, we performed two rounds of FACS on a library of pooled transformants to isolate YZy505, which produces 681 mg/L ± 29 mg/L isobutanol in dark fermentations from 2% glucose, 20-times higher the starting strain (YZy480, Fig. 6b). Thus, our biosensor can assist in the construction of mitochondrial or cytosolic BCHA pathways, and is effective at identifying high-producing strains, including optogenetically controlled strains for enhanced flux towards isobutanol production.

## Discussion

Genetically encoded biosensors have been developed to monitor microbial biosynthesis of several products of interest[51–60].

However, to our knowledge, biosensors for specific BCAAs and BCAA-derived products have not been reported in *S. cerevisiae*, which could enable high-throughput screens to develop strains for their production. Here we show that the endogenous Leu3p regulator of BCAA metabolism can be harnessed to develop genetically encoded biosensors for BCAA-derived product biosynthesis, such as the BCHAs isobutanol and isopentanol. Each application we describe demonstrates the use of the biosensor in either the isobutanol or isopentanol configuration for a specific purpose: to isolate high-producing strains from libraries of combinatorially assembled mitochondrial pathways (Fig. 2); to identify hyperactive enzymes for mitochondrial production of isobutanol (Fig. 3); or production of isopentanol across both mitochondria and cytosol (Fig. 4); to aid in the construction of a cytosolic isobutanol pathway (Fig. 5); and to use it in combination with optogenetic actuators, which opens future potential applications in dynamic control of metabolism (Fig. 6).

The fact that the biosensor acts by monitoring the intracellular concentrations of α-IPM[21,22] offers several advantages. Because α-IPM is a by-product of isobutanol and a precursor of isopentanol, the biosensor can be designed in two different configurations that indicate relative production of each BCHA specifically (Fig. 1 and Supplementary Fig. 4). The high selectivity stemming from the Leu3p mechanism of activation also makes each biosensor configuration unlikely to cross-react with other products, in contrast to previously reported alcohol biosensors in *E. coli*[61,62] and *S. cerevisiae*[63]. Furthermore, by monitoring the concentration of an intracellular metabolite (α-IPM), as opposed to the concentration of a secreted end product that easily traverses cell membranes bi-directionally, Leu3p-dependent biosensors facilitate the detection of BCAA-derived product biosynthesis in each individual cell (using flow cytometry), rather than the average production in a fermentation.

To improve isobutanol or isopentanol production in yeast, biosynthetic pathways have been engineered in mitochondria[29,37], cytosol[38,39], or simultaneously in both compartments[37,39,41,64], which makes the versatility of the biosensor to detect BCHA production in both compartments particularly valuable. For the mitochondrial isobutanol pathway, the isobutanol configuration of the biosensor allowed us to find strains from combinatorial libraries that produce, on average, more than twice as much isobutanol as strains randomly selected without assistance of a biosensor (Fig. 2), as well as strains with enhanced ALS activity (Ilv2p/Ilv6p, Fig. 3). For the cytosolic pathway, the same biosensor configuration helped us open two key bottlenecks at the KARI (*Ec_ilvC*^P2D1-A1, Supplementary Fig. 12), and DHAD (*Ll_IlvD*^I433V, Fig. 5c) metabolic steps. To our knowledge, no DHAD mutant to enhance isobutanol production has been previously reported. Additionally, the isopentanol configuration of the biosensor allowed us to identify strains with enhanced (α-IPM) synthase activity (Leu4p, Fig. 4), which is present in both mitochondria and cytosol. Altogether, high-throughput screens enabled by the biosensor in both configurations, allowed us to improve both mitochondrial and cytosolic isobutanol production by as much as 15- and 9-fold, respectively, as well as isopentanol production by approximately 20-fold (relative to wild-type strains).

In all our demonstrations, strains with higher fluorescence isolated after three rounds of sorting always have increased BCHA production, suggesting our biosensor has a very low rate of false positives. Beyond our initial characterizations (Fig. 1b, d), analyzing the strains randomly picked from subsequent sorting experiments confirmed the strong linear correlation ($R^2 = 0.95$, 0.96, 0.95) between fluorescence readout and BCHA production in actual screens (Supplementary Fig. 15), identifying virtually no false positives. This high efficiency is likely due to the fact that the source of diversity in all our applications is upstream of the α-IPM intermediate, for which the biosensor was designed. While most of the biosynthetic pathway and previously identified bottlenecks lay upstream of α-IPM, if the downstream pathway became limiting, the rate of false positives for isobutanol production could increase. Similarly, we do not expect the same high-throughput screens used in this study to be effective at identifying enhanced enzymes downstream of α-IPM. However, sorting strains with decreased fluorescence, reflecting increased rates of α-IPM processing, might be helpful in these scenarios. Furthermore, screening for random mutations in the genome could also lead to higher rates of false positives. Future research will determine how the biosensor operates in these specific settings.

This biosensor is applicable to several lines of research in biotechnology, basic science, and metabolic control. Their most immediate application will likely be to accelerate the development of strains for the production of isobutanol and isopentanol as promising advanced biofuels. However, because the biosensor is controlled by the activity of the BCAA regulator Leu3p, rather than directly sensing isobutanol or isopentanol concentrations, it may also be useful to develop strains for the production of valine, leucine, or other products derived from their metabolism[1–8]. An advantage of biosensors based on transcription factors is that they can be used to control the expression of any gene of interest. Therefore, Leu3p can be used to control the expression of not only reporter genes to develop high-throughput screens for strains with enhanced production (as demonstrated in this study), but also of essential or antibiotic resistance genes to develop high-throughput selection assays[54,62,65,66], or of key metabolic genes to develop autoregulatory mechanisms for dynamic control of product biosynthesis[18,67]. In contrast, the BCAA biosensor previously reported in mammalian cells[68,69] is based on fluorescence resonance energy transfer (FRET) and only measures the collective intracellular levels of valine, leucine, and isoleucine, which limits its applications.

Additionally, because BCAA biosynthesis initiates in mitochondria, our biosensor is potentially useful to study fundamental questions, not only in BCAA metabolism, but also in mitochondrial biology. While previously reported biosensors based on GFP-ligand-binding-protein chimeras, FRET, or its bioluminescence equivalent (BRET) have been developed to probe the mitochondrial metabolic state[70], our biosensor is capable of probing the metabolic activity of a mitochondrial biosynthetic pathway. Finally, the fact that the biosensor is functional in strains carrying optogenetic controls (Fig. 6) raises the intriguing possibility of simultaneously monitoring and controlling production of BCAAs and BCAA-derived products, which may open new research avenues in closed loop metabolic control and cybergenetics for metabolic engineering[71].

## Methods

**General molecular biology techniques**. Genomic DNA extractions were carried out using the standard phenol chloroform procedure[72]. Genotyping PCRs were performed using GoTaq Green Master Mix (Promega, Madison, WI, USA). The oligonucleotides used in this study (Supplementary Table 9) were obtained from Integrated DNA Technologies (IDT, Coralville, IA, USA). *Escherichia coli* DH5α was used for routine transformations. All of the constructed plasmids were verified by DNA sequencing (GENEWIZ, South Plainfield, NJ, USA).

**Growth media**. Unless otherwise specified, yeast cells were grown at 30 °C on either YPD medium (10 g/L yeast extract, 20 g/L peptone, 0.15 g/L tryptophan and 20 g/L glucose) or synthetic complete (SC) drop out medium (20 g/L glucose, 1.5 g/L yeast nitrogen base without amino acids or ammonium sulfate, 5 g/L ammonium sulfate, 36 mg/L inositol, and 2 g/L amino acid drop out mixture) supplemented with 2% glucose, or non-fermentable carbon sources such as 2% galactose, or the mixture of 3% glycerol and 2.5% ethanol. 2% Bacto™-agar (BD, Franklin Lakes, NJ, USA) was added to make agar plates.

**Assembly of DNA constructs**. DNA construction was performed using standard restriction-enzyme digestion and ligation cloning and isothermal assembly (Gibson Assembly)[73]. Endogenous *S. cerevisiae* genes (*ILV6*, *LEU4*, and *AFT1*) were amplified from genomic DNA of CEN.PK2-1C by PCR using a forward primer containing an NheI restriction recognition site and a reverse primer containing an XhoI restriction recognition site (Supplementary Table 9). This enabled subcloning of PCR-amplified genes into pJLA vectors[29] and pJLA vector-compatible plasmids. Genes from other organisms, including *Bs_alsS*, *Ec_ilvC*$^{P2D1-A1}$ and *Ll_ilvD* were codon optimized for *S. cerevisiae* and synthesized by Bio Basic Inc. (Amherst, NY, USA). These genes were designed with flanking NheI and XhoI sites at the 5′ and 3′ ends, respectively. All plasmids constructed or used in this study are listed in Supplementary Table 2.

*Constructing the isobutanol configuration of the biosensor*. The reporter for the biosensor in its isobutanol configuration was constructed by placing the *S. cerevisiae LEU1* promoter (P$_{LEU1}$), in front of the yeast-enhanced green fluorescent protein (yEGFP) fused to a PEST tag. The *LEU1* promoter and yEGFP-PEST fragments amplified via PCR were inserted by Gibson assembly into the vector JLAb131 to make an intermediate plasmid (pYZ13) containing the fragment P$_{LEU1}$-yEGFP_PEST-T$_{ADH1}$, which was then subcloned into a *HIS3* locus integration (His3INT) vector pYZ12B[46], yielding the intermediate plasmid pYZ14. The truncated *LEU4*$^{1-410}$ was amplified from genomic DNA of CEN.PK2-1C and subcloned into plasmid JLAb23, which contains the constitutive promoter P$_{TPI1}$ and terminator T$_{PGK1}$. The P$_{TPI1}$-*LEU4*$^{1-410}$-T$_{PGK1}$ cassette from the resulting plasmid pYZ2 was then subcloned into pYZ14 using sequential gene insertion cloning[29] to form the plasmid pYZ16 (Supplementary Table 2).

*Constructing the isopentanol configuration of the biosensor*. To construct the biosensor in its isopentanol configuration, we removed the PEST-tag of the yEGFP reporter in plasmid pYZ14 by annealed oligo cloning. The two single-stranded overlapping oligonucleotides (Yfz_Oli59 and Yfz_Oli60) were annealed and cloned directly into the overhangs generated by restriction digest of pYZ14 at SalI and BsrGI sites. The resulting plasmid (pYZ24) contains cassette P$_{LEU1}$-yEGFP-T$_{ADH1}$. Next, we added a catalytically active and leucine-insensitive *LEU4* mutant (*LEU4*$^{ΔS547}$, a deletion in Ser547)[28,37] to pYZ24 to form the isopentanol configuration of the biosensor, pYZ25. The deletion in Ser547 was achieved by site-directed mutagenesis using a plasmid containing wild-type *LEU4* (*LEU4*$^{WT}$). *LEU4*$^{ΔS547}$ was then subcloned into JLAb23 to form plasmid pYZ1. The P$_{TPI1}$-*LEU4*$^{ΔS547}$-T$_{PGK1}$ cassette from pYZ1 was then subcloned into pYZ24 to form the plasmid pYZ25.

*Constructing template plasmids for error-prone PCR*. Plasmid pYZ125, used to prepare the random mutagenesis libraries, was constructed by subcloning the region of JLAb131 spanning from the *TDH3* promoter (P$_{TDH3}$) to the *ADH1* terminator (T$_{ADH1}$) into the CEN plasmid pRS416 to generate pYZ125[74]. *ILV6*, *LEU4*, and *Ll_ilvD* were subcloned into pYZ125, after linearization with NheI and XhoI, to create pYZ127, pYZ149, and pYZ126, respectively. The *ILV6*$^{V90D/L91F}$ mutant was generated using QuikChange site-directed mutagenesis and subcloned into pYZ125 to create pYZ148[35]. Leucine inhibition insensitive *LEU4* mutants *LEU4*$^{ΔS547}$ and *LEU4*$^{1-410}$ were subcloned into pYZ125 to form pYZ154 and pYZ155, respectively (Supplementary Table 2).

*Constructing δ-integration cassettes*. We used a previously developed δ-integration (δ-INT) vector, pYZ23[46], to integrate multiple copies of gene cassettes into genomic YARCdelta5 δ-sites, the 337 bp long-terminal-repeat of *S. cerevisiae* Ty1 retrotransposons (YARCTy1-1, SGD ID: S000006792). The selection marker in pYZ23 is the shBleMX6 gene, which encodes a protein conferring resistance to zeocin and allows selection of different numbers of integration events by varying zeocin concentrations[75]. Resistance to higher concentrations of zeocin correlates with a higher number of gene cassettes integrated into δ-sites. To construct δ-integration plasmids pYZ33, pYZ113, and pYZ34, we used restriction site pairs XmaI/AscI (to extract gene cassettes) and MreI/AscI (to open pYZ23)[29,76]. Plasmid pYZ417 was constructed by sequentially inserting cassettes P$_{TDH3}$-*Ec_ilvC*$^{P2D1-A1}$-T$_{CYC1}$ (from pYZ196), P$_{TEF1}$-*Ll_ilvD*$^{I433V}$-T$_{TPS1}$ (from pYZ383), and P$_{Gal1-S}$-*Bs_alsS*-T$_{ACT1}$ (from pYZ384) into the δ-integration plasmid EZ-L235[46], which contains the OptoEXP-PDC1 cassette (P$_{C120}$-*PDC1*-T$_{ACT1}$). All of the δ-integration plasmids were linearized with PmeI prior to yeast transformation.

**Error-prone PCR and random mutagenesis library construction**. The random mutagenesis libraries of *ILV6*, *LEU4* and *Ll_ilvD* were generated by error-prone PCR using the GeneMorph II Random Mutagenesis Kit (Agilent Technologies, Santa Clara, CA, USA). The CEN plasmids harboring wild-type *ILV6* (pYZ127), *LEU4* (pYZ149), *Ll_ilvD* (pYZ126) or the *ILV6*$^{V90D/L91F}$ mutant (pYZ148) were used as templates for error-prone PCR. Various amounts of DNA template were used in the amplification reactions to obtain low (0–4.5 mutations/kb), medium (4.5–9 mutations/kb), and high (9–16 mutations/kb) mutation frequencies as described in the product manual. Primers Yfz_Oli198 and Yfz_Oli242, which contain NheI and XhoI restriction sites at their 5′ ends, respectively, were used to amplify and introduce random mutations into the region between the start codon

and stop codon of each gene. The PCR fragments were incubated with DpnI for 2 h at 37 °C to degrade the template plasmids before purification with a the QIAquick PCR purification kit (Qiagen). The purified PCR fragments were digested overnight at 37 °C using NheI and XhoI, and ligated overnight at 16 °C with pYZ125. The same process was used to create an ep_library of the leucine regulatory domain of Leu4p, except the forward primer contained a BglII restriction site in its 5′ end, rather than an NheI site. The plasmid libraries that resulted were transformed into ultra-competent *E. coli* DH5α and plated onto LB-agar plates (five 150 mm petri dishes per library) containing 100 μg/ml of ampicillin. After incubating the plates overnight at 37 °C, the resulting lawns (with a library size of ~10$^9$) were scraped off the agar plates and the plasmid libraries were extracted using a QIAprep Spin Miniprep Kit (QIAGEN). The plasmid libraries were subsequently used for yeast transformation. Based on yeast transformation efficiency with these plasmid libraries, and the number of transformants collected for FACS, we estimate the size of our pre-sorted libraries to be ~10$^6$–10$^7$ variants.

**Yeast strains and yeast transformations**. *S. cerevisiae* strain CEN.PK2-1C (*MATα ura3–52 trp1-289 leu2-3,112 his3-1 MAL2-8c SUC2*) and its derivatives were used in this study (Supplementary Table 1). Deletions of *BAT1*, *BAT2*, *ILV6*, *LEU4*, *LEU9*, *ILV3*, *TMA29*, *PDC1*, *PDC5*, *PDC6*, and *GAL80* were obtained using PCR-based homologous recombination. DNA fragments containing lox71- and lox66-flanked antibiotic resistance cassettes were amplified with PCR from plasmids containing an antibiotic resistance gene, using primers with 50–70 base pairs of homology to regions upstream and downstream of the ORF targeted for deletion[30]. Transformation of the gel-purified PCR fragments was performed using the lithium acetate method[77]. Cells transformed using antibiotic resistance markers, were first plated onto nonselective YPD plates for overnight growth, then replica plated onto YPD plates containing 300 μg/mL hygromycin (Invitrogen, Carlsbad, CA), 200 μg/mL nourseothricin (WERNER BioAgents, Jena, Germany), or 200 μg/mL Geneticin (G-418 sulfate, Gibco, Life Technologies, Grand Island, NY, USA). Chromosomal integrations and plasmid transformations were also performed using the lithium acetate method[77]. Cells transformed for δ-integration were first incubated in YPD liquid medium for six hours and then plated onto nonselective YPD agar plates for overnight growth. The next day, cells were replica plated onto YPD agar plates containing 300 μg/mL (for FACS) or higher concentrations (500, 800, 1200, and 1500 μg/mL for titrating higher copy numbers of integrations) of Zeocin (Invitrogen, Carlsbad, CA), and incubated at 30 °C until colonies appeared. Cells transformed with plasmid libraries or 2μ plasmids (used for FACS) were plated on three 150 mm petri plates per library. The resulting lawns (containing ~6.5 × 10$^4$ individual transformants in each plate, for a total of ~2 × 10$^5$ transformants) were scraped off the agar plates, small aliquots of the collected cells were then grown to exponential phase in fresh medium, and used for flow cytometry and FACS (see below). All strains with gene deletions or chromosomal integrations (expect δ-integrations) were genotyped with positive and negative controls to confirm the removal of the ORF of interest or the presence of integrated DNA cassette.

*Strains with cytosolic isobutanol pathway*. We first constructed a baseline strain for cytosolic isobutanol production (YZy449). We used constitutive promoters to express *Bs_alsS*, *Ec_ilvC*$^{P2D1-A1}$, and *AFT1*, and the galactose-inducible and glucose-repressible promoter P$_{GAL10}$ to express *Ll_ilvD*. This approach allows us to use the same strain to screen for *Ec_ilvC*$^{P2D1-A1}$ and *Ll_ilvD* mutants using different carbon sources (galactose and glucose, respectively). We first deleted *ILV3* and *TMA29* in strain YZy121, which contains the isobutanol-configured biosensor, resulting in strain YZy443. We then transformed strain YZy443 with the *URA3* integration cassette from plasmid pYZ196 (loxP-*URA3*-loxP-P$_{TDH3}$-*AFT1*-T$_{ADH1}$-[P$_{TEF1}$-*Bs_alsS*-T$_{ACT1}$-P$_{TDH3}$-*Ec_ilvC*$^{P2D1-A1}$-T$_{ADH1}$]-P$_{GAL10}$-*Ll_ilvD*-T$_{ACT1}$), resulting in strain YZy447. Next, we recycled the *URA3* marker using the Cre-loxP site-specific recombination system[78] and counter selected on YPD plates with 1 mg/mL of 5-FOA (see below), resulting in the final strain YZy449. Both YZy447 and YZy449 make ~170 mg/L of isobutanol in fermentations using galactose as carbon source but are unable to make isobutanol from glucose (Supplementary Fig. 16).

*Applying the biosensor in optogenetically controlled strains*. To combine the biosensor (in its isobutanol configuration) with optogenetic controls of the cytosolic isobutanol pathway, we integrated OptoINVRT7[49] (using EZ-L439) into the *HIS3* locus of YZy90, a *gal80Δ*, triple *pdcΔ* (*pdc1Δ pdc5Δ pdc6Δ*) strain containing a constitutively expressed copy of *PDC1* in a 2μ plasmid, resulting in YZy480. We removed the 2μ-*PDC1* plasmid using 5-FOA (see below) to generate strain YZy481, which is able to grow in medium supplemented with glycerol and ethanol, but not in medium supplemented with glucose. We next introduced the biosensor in its isobutanol configuration into the *GAL80* locus of YZy481 to yield YZy487, and used GFP fluorescence intensity measurements and genotyping to confirm the biosensor was successfully integrated. A dark-inducible cytosolic isobutanol pathway and the light-inducible *PDC1* were then introduced to strain YZy487 via δ-integration. After two rounds of FACS, we analyzed the GFP fluorescence and isobutanol titers of ten colonies in dark fermentations with 2% glucose (see below). The colony with the highest GFP fluorescence intensity, corresponding to the highest isobutanol titer, was chosen as the host strain (YZy502) for further

enhancement of the cytosolic isobutanol production. To achieve a strain with even higher metabolic flux through the cytosolic isobutanol pathway, we introduced a 2μ plasmid pYZ350, containing partial cytosolic isobutanol pathway genes, and applied FACS to isolate high-producing transformants.

**Flow cytometry/FACS**. A BD LSRII Multi-Laser flow cytometer equipped with FACSDiva software V.8.0.2 (BD Biosciences, San Jose, CA) was used to quantify yEGFP fluorescence at an excitation wavelength of 488 nm and an emission wavelength of 510 nm (525/50 nm bandpass filter). Cells were gated on forward scatter (FSC) and side scatter (SSC) signals to discard debris and probable cell aggregates (Supplementary Fig. 17). The typical sample size was 50,000 events per measurement. A BD FACSAria Fusion flow cytometer with FACSDiva software was used for fluorescence activated cell sorting (FACS) with a 488 nm excitation wavelength and a bandpass filter of 530/30 for yEGFP detection. Cells exhibiting high levels of GFP fluorescence (top ~1%) were sorted. Sorted cells were collected into 1 mL of medium (see below), and 50 μL of each sample of collected sorted were streaked on a corresponding agar plate and incubated at 30 °C to obtain 24 random colonies. For each of the single colonies isolated, MFI and BCHA production were measured. The rest of the sorted cells (~950 μL) were incubated at 30 °C and at 200 rpm agitation to reach the stationary growth phase, followed by subculturing (1:100 dilution) in the same medium for the next round of sorting. FlowJo X software (BD Biosciences, San Jose, CA) was used to analyze the flow cytometry data.

*Sample preparation for flow cytometry assays and FACS high-throughput screens*. Fluorescence measurements and FACS were performed on samples in mid-exponential growth phase. Single colonies from agar plates or yeast transformation libraries diluted 1:100 were first cultured overnight until stationary phase in synthetic complete (SC), or synthetic complete minus uracil (SC-ura) medium, supplemented with 2% glucose or galactose (for screening the library of strains containing varying copies of $Ec\_ilvC^{P2D1-A1}$). Overnight cultures were diluted 1:100 in the same fresh medium and grown to mid-exponential growth phase (12–13 h after inoculation). The growth media used for flow cytometry assays and FACS of *ILV6*-samples contain four times more valine (2.4 mM) than the synthetic defined medium described above. For strains with *PDC1* and *Bs_alsS* controlled by optogenetic circuits (*OptoEXP*[46] and *OptoINVRT7*[49], respectively), cultures were incubated for 8 h under constant blue light after inoculation with cells in stationary phase, followed by 10 h of incubation in the dark (see below) before flow cytometry or FACS. For all flow cytometry assays, cultures were diluted to an $OD_{600}$ of approximately 0.1 with PBS. All samples used for FACS were diluted in fresh medium identical to their growth medium to $OD_{600}$ of approximately 0.8.

**Measurement of the response of the biosensor to α-IPM, α-KIV, and α-K3MV**. Measurements of the responses of the biosensor to α-IPM, α-KIV and α-K3MV were carried out in a CEN.PK2-1C background strain, with the biosensor in its isobutanol configuration integrated at the *HIS3* locus (YZy121). A single colony from an agar plate was cultured overnight in SC medium supplemented with 2% glucose. The overnight culture was diluted 1:100 in fresh SC medium supplemented with 2% glucose, and 1 mL of the diluted culture was added to each well of a 24-well cell culture plate (Cat. 229524, CELLTREAT Scientific Products, Pepperell, MA, USA). Then, 10 μL of freshly made α-IPM (pH 7.0), α-KIV (pH 7.0), or α-K3MV (pH 7.0) solutions were added to each well to reach different final concentrations ranging from 0 μM to 75 μM. The 24-well plate was shaken for 12 h in an orbital shaker (Eppendorf, New Brunswick, USA) at 30 °C and at 200 rpm agitation. The GFP fluorescence of each sample in exponential phase ($OD_{600} = 0.8$–1.2) was measured using flow cytometry, with samples diluted in PBS in plates (50 μL of culture into 200 μL of buffer) for α-IPM and α-KIV, or in tubes (1 mL into 1 mL) for α-K3MV. The diluted samples were then kept on ice and until flow cytometry measurements were performed.

**Yeast fermentations for isobutanol or isopentanol production**. High-cell-density fermentations were carried out in sterile 24-well microtiter plates (Cat. 229524, CELLTREAT Scientific Products, Pepperell, MA, USA) in an orbital shaker (Eppendorf, New Brunswick, USA) at 30 °C and at 200 rpm agitation. Single colonies were grown overnight in 1 mL of synthetic complete (SC), or synthetic complete minus uracil (SC-ura) medium, supplemented with 2% glucose. The next day, 10 μL of the overnight culture were used to inoculate 1 mL of SC (or SC-ura) medium supplemented with 2% glucose in a new 24-well plate. After 20 h, the plates were centrifuged at 234 *g* for 5 min, the supernatant was discarded, and cells were resuspended in 1 mL of SC (or SC-ura) supplemented with 15% glucose (or galactose). The plates were covered with sterile adhesive SealPlate® films (Cat. # STR-SEAL-PLT; Excel Scientific, Victorville, CA) and incubated for 48 h at 30 °C with 200 rpm shaking. The SealPlate® films were used in all 24-well plate fermentations to maintain semi-aerobic conditions in each well, and to prevent evaporation and cross-contamination between wells. At the end of the fermentations, the $OD_{600}$ of the culture in each well was measured in a TECAN infinite M200PRO plate reader (Tecan Group Ltd., Männedorf, Switzerland). Plates were then centrifuged for 5 min at 234 *g*, and the supernatant from each well was analyzed using HPLC as described below.

Fermentations of strains with optogenetic controls were also carried out in sterile 24-well microtiter plates, as described above, but with modifications as previously described[46] (see Supplementary Note 7). The parameter ρ, the cell density at which we switched cells from growing in blue light to dark and θ, the incubation time in the dark before fermentation used in the initial screening fermentations are an $OD_{600}$ of 6 and 6 h, respectively, which were estimated based on the characterization and modeling of the optogenetic circuits OptoEXP[46] and INVRT7[49]. The optimal ρ ($OD_{600}$ of 5) and θ (6 h) of the best isobutanol-producing strains (YZy502 and YZy505) were determined experimentally by measuring the isobutanol titers from fermentations using different ρ and θ values. To vary these parameters, a single colony of each strain was first grown to stationary phase in SC (for YZy502) or SC-ura (for YZy505) medium supplemented with 2% glucose under constant blue light at 30 °C. The overnight cultures were diluted to an $OD_{600}$ of 0.1 in the same medium. We began incubations (at 30 °C and 200 rpm) of the diluted cultures at different times and under pulsed blue light to achieve cultures with different $OD_{600}$ values, which correspond to variations in ρ, ranging from 1 to 9. After measuring the $OD_{600}$ of each culture, we switched them from light to dark and incubated for 6 h at 30 °C and 200 rpm. After the dark incubation period, the cells were centrifuged and resuspended in 1 mL of the same medium supplemented with 2% glucose. The plates were covered with sterile adhesive SealPlate® films (Cat. # STR-SEAL-PLT; Excel Scientific, Victorville, CA) and incubated in the dark (wrapped in aluminum foil) for 48 h at 30 °C and 200 rpm. Subsequently, the cultures were prepared for HPLC analysis (see below). To determine the optimal θ, we diluted the overnight culture to an $OD_{600}$ of 0.1 and incubated it at 30 °C and 200 rpm, and under pulsed blue light to reach an $OD_{600}$ of 5, which was the optimal ρ determined in the previous experiment. Next, we incubated the cells in the dark (30 °C and 200 rpm) for different numbers of hours, ranging from 1 to 10, which correspond to variations in θ. After the dark incubation period, cells were centrifuged and resuspended in 1 mL of the same medium supplemented with 2% glucose, followed by 48 h of incubation in the dark (30 °C and 200 rpm) and HPLC analysis as described below.

**Analysis of biosensor fluorescence, intracellular α-IPM, and BCHA production throughout low-cell density fermentations**. Low and high-isobutanol (YZy121, YZy235) or isopentanol producers (SHy187, SHy159) were grown overnight in 1 mL of SC (or SC-ura) medium supplemented with 2% glucose in 24-well plates at 30 °C. 100 μL of overnight culture was used to inoculate 100 mL of SC (or SC-ura) medium supplemented with 15% glucose at 30 °C, and agitated at 200 rpm in an orbital shaker for a time course of 30 h. Timepoints were taken at 5 h, 12 h, and 30 h. For each time point, samples were collected for analysis of either cellular florescence, intracellular α-IPM, or extracellular BCHA concentration. Cellular florescence was measured via flow cytometry as described above. Intracellular α-IPM concentration was measured with Thermo Fisher Q Exactive HPLC-Orbitrap MS equipment. The protocol for metabolite extraction from cells was based on previously described methods[60] with minor modifications. At each time point, an extraction solution containing 40:40:20 (v/v/v) methanol:acetonitrile:ddH$_2$O with 0.5% formic acid stored at −20 °C, was used to extract metabolites over a volume of cells equivalent to 15 mL of culture at an $OD_{600}$ of 1. Although volumes collected for each time point differ, equal cell masses were pelleted by centrifuging at 2107 *g* for 10 min at 4 °C. The cell pellet was vortexed with 1 mL of extraction solution and incubated for 1 min at room temperature. A total of 88 μL of 15.8% (w/v) NH$_4$HCO$_3$ was added immediately after, and mixed by vortex to neutralize the extraction solution. The resulting mixture was incubated at −20 °C for 15 min and centrifuged at 17,000 *g* for 8 min at 4 °C. A total of 150 μL of supernatant was dried via vacufuge for 2 h. The dried metabolites were resuspended in 50 μL of ddH$_2$O and analyzed using an Atlantis T3 3 μm 2.1 × 150 mm reversed-phase column (Waters, Part No. 186003716, Milford, MA, USA). Gas chromatography (GC/MS) was used to measure extracellular product concentration of isobutanol or isopentanol. A total of 800 μL of the fermentation broth was centrifuged at 17,000 *g* for 40 min at 4 °C to remove cells and residual debris. The supernatant was subjected to an alcohol extraction with hexane, in which hexane and supernatant were mixed at a 1:1 ratio, vortexed for 15 min, and then centrifuged at 17,000 *g* for 10 min at 4 °C. The organic phase was analyzed using a DB-WAX UI 0.5 μm gas chromatography column (Agilent, Part No. 122-7033UI, Santa Clara, CA, USA).

**Digital droplet PCR to estimate library and sorted strain genotypes**. The Digital Droplet PCR (ddPCR) experiment was performed on a Bio-Rad QX200 Droplet Digital PCR system with an Automated Droplet Generator using QX200 ddPCR Evagreen Supermix according to the manufacturer's instructions. Four PCR primers were used (Supplementary Table 9) to amplify a genomic single copy reference, *ILV2*, *ARO10*, and the Zeocin resistance cassette on genomic DNA templates from the pre-sorted library (PSL) as well as from the high-producers isolated from the second round of FACS in Fig. 2 (Y436-439, Y442 and Y443). The copy numbers of each amplification target were normalized to the counts of the single copy genome reference. As the upstream (A) and downstream pathway (B) share homology with the complete pathway (C), the following linear equations were solved to determine the number of integrations of each cassette, where X is the copy number of the zeocin resistance cassette, Y is the copy number of *ILV2*,

and Z is the copy number of *ARO10*:

$$A + B + C = X \quad (1)$$

$$A + C = Y \quad (2)$$

$$B + C + 1 = Z \quad (3)$$

As *ARO10* has a single integration in wild-type yeast, 1 copy is added to Eq. (3). The copies of unique strains are rounded to the nearest integer, while copies in PSL, are reported as the number of average integrations normalized per genome in the diverse library population.

**Removal of plasmids from *Saccharomyces cerevisiae*.** *URA3* plasmids in yeast strains were removed by 5-fluoroorotic acid (5-FOA) selection[79]. Cells were first grown in YPD overnight and then streaked on SC agar plates containing 1 mg/mL 5-FOA (Zymo Research, Orange, CA, USA). A single colony from an SC/5-FOA agar plate was streaked again on a new SC/5-FOA agar plate. To confirm that strains were cured of the *URA3*-containing plasmid, they were inoculated into SC-ura medium supplemented with 2% glucose; strains lacking the *URA3*-containing plasmid were not able to grow in SC-ura medium.

**Yeast plasmid isolation.** Plasmid isolation from yeast was performed according to a user-developed protocol from Michael Jones (protocol PR04, Isolation of plasmid DNA from yeast, QIAGEN) using a QIAprep Spin Miniprep kit (QIAGEN, Valencia, CA, USA)[80]. The isolated plasmids were retransformed into *E. coli* DH5α to produce plasmids at higher titer for subsequent sequencing and retransformation into the parental yeast strain.

**Analysis of BCHA production.** The concentrations of isobutanol, and isopentanol were determined with high-performance liquid chromatography (HPLC) using an Agilent 1260 Infinity instrument (Agilent Technologies, Santa Clara, CA, USA). Samples were centrifuged at 17,000 *g* for 40 min at 4 °C to remove residual cells and other solid debris, and analyzed using an Aminex HPX-87H ion-exchange column (Bio-Rad, Hercules, CA USA). The column was eluted with a mobile phase of 5 mM sulfuric acid at 55 °C and with a flow rate of 0.6 mL/min for 50 min. The chemical concentrations were monitored with a refractive index detector (RID) and quantified by comparing the peak areas to those of standards with known concentrations.

**Statistical analysis.** Two-sided Student's *t* tests were performed using GraphPad Prism (version 8.0 for Mac OS, GraphPad Software, San Diego, California USA, www.graphpad.com) to determine the statistical significance of differences observed in product titers between strains. Probabilities (*P*-values) less than (or equal to) 0.05 are considered sufficient to reject the null hypothesis (that the means of the two samples are the same) and accept the alternative hypothesis (that the means of the two samples are different).

**Reporting summary.** Further information on research design is available in the Nature Research Reporting Summary linked to this article.

## Data availability
The authors declare that all data supporting the findings of this study are available within the paper (and its Supplementary Information files). The source data underlying Figs. 1–6, Supplementary Figs. S2–7, S9, S10, S12, S13, S15, S16, and Supplementary Table 3 are provided in the Source Data File.

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

## Acknowledgements

The authors thank Christina DeCoste and Katherine Rittenbach, who assisted with all flow cytometry instrumentation. We also thank Wenyun Lu and Joshua D. Rabinowitz, who generously provided equipment and assistance in measuring intracellular α-IPM metabolite concentrations. This work was supported by the U.S. Department of Energy, Office of Science, Office of Biological and Environmental Research, Genomic Science Program under award number DE-SC0019363, as well as by the National Science Foundation, and NSF CAREER Award CBET-1751840 (to J.L.A.). This work was also supported by the NSF Graduate Research Fellowship Program grant DGE-1656466, the P.E.O. Scholar Award, and the Harold W. Dodds Fellowship from Princeton University (to S.K.H.), as well as by the NIGMS of the National Institutes of Health under grant number T32GM007388 (to J.D.C.). The content of this article is solely the responsibility of the authors and does not necessarily represent the official views of the National Institutes of Health. J.L.A. is also supported by The Pew Charitable Trusts, The Camille Dreyfus Teacher-Scholar Award, The Eric and Wendy Schmidt Transformative Technology Fund, and grants from Princeton University and the Andlinger Center for Energy and the Environment.

## Author contributions

Conceptualization and methodology: Y.Z., J.D.C., S.K.H., and J.L.A.; investigation: Y.Z., J.D.C., S.K.H., C.C.L., S.A.G.E., J.B.W., W.W., and J.L.A.; writing of original draft: Y.Z., S.K.H., and J.L.A.; review and editing: all authors; funding acquisition: J.L.A.; supervision: J.L.A.

## Competing interests

J.L.A. and the Whitehead Instituted/MIT have submitted a patent application to the US patent office pertaining some elements of the biosensor design described in this study (US20160326535A1). The remaining authors declare no competing interests.

## Additional information

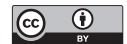

