## [Peer Review File · Nature Communications]

Reviewers' Comments:

Reviewer #1:

Remarks to the Author:

The manuscript of Zhang et al. describes an impressive and interesting work in industrial biotechnology with yeast. The authors developed a biosensor for an intermediate of branched-chain amino acid metabolism (IPM) and used the biosensor in two different set-ups to perform high-throughput screens for improved isobutanol and isopentanol production. The biosensor turned out to be quite effective for this. It will be extremely valuable to further improve isobutanol and isopentanol production with yeast, two very promising biofuels.

I have only some minor comments:

- 1) The authors claim that they have developed two biosensors. However, it is rather only one biosensor which is used in two different set-ups.
- 2) L. 134: it should be explained why LEU2 was deleted and not LEU1 which actually encodes the enzyme converting IPM. Otherwise it is confusing.
- 3) The authors several times stress that it is very important that their new biosensor can be used to optimize as well a mitochondrial as also a cytosolic isobutanol pathway. Why? This is obvious. The biosensor actually is completely independent from intracellular localization of the pathway, especially as also the intermediates of branched-chain amino acid metabolism seem to be permeable to membranes. Therefore, this statement seems to be unnecessary and confusing.
- 4) The authors do not discuss the limits of their new biosensor. As it detects an intermediate in metabolism it is only useful to screen for improvements in the upstream part of the pathway. If the downstream Ehrlich pathway would be or would become limiting the sensor could not be used any longer. Or can it be used then to detect a decrease in IPM concentrations? This should be discussed.
- 5) L. 581-585: the conclusion is not correct. The authors showed on page 8 (line 164-165) that extracellular addition of KIV and IPM stimulated the biosensor, demonstrating permeability. Therefore, also these intermediate are permeable and could traverse membranes.
- 6) 1500% and 2000% should be replaced with 15-fold and 20-fold.

Reviewer #2:

Remarks to the Author:

In this manuscript, Zhang and coworkers present the design and construction two biosensors for the indirect detection of isobutanol and isopentanol in yeast. Following, several examples are presented in which the biosensors were used in combination with FACS to isolate isobutanol producers or to engineer enzymes (relieve from feedback-inhibition; increased enzymatic activity) involved in BCAA-synthesis. Finally, a mitochondrial- and a cytosolic pathway for isobutanol are presented.

The manuscript is well written, but way too long. In particular, the introduction and the discussion could be substantially shorter. In addition, one or even two application examples could be removed from the text, e.g. the isolation of isobutanol-producing strains as this is a simple enrichment. Unfortunately, the abstract promises much more than the manuscript can keep: The Leu3p-based transcriptional biosensor does not detect any of the desired products but only a byproduct or intermediate (depending on the product, see major remarks below). In addition, the biosensor characterization is not very detailed as the operational range is only partially investigated the dynamic range is not discussed at all. In addition, some important experiments such as intracellular quantification of α -IPM in the different biosensor-setups are also missing (see below). Taken together, I think that this manuscript is more appropriate for a journal such as ACS Synthetic Biology.

Major remarks:

In general, I find the term "isobutanol biosensor" (e.g. line 128) or "isopentanol biosensor" very misleading as either a byproduct or a pathway intermediate is detected. In addition, except for the PEST-tag the sensors are essentially the same. It is the pathway configuration, which does the trick. In addition, it is great to see that engineered strains give a good biosensor response, but when only a byproduct/intermediate and not the desired product is detected, a large number of false positive variants (no increased product titer) can be expected in a real-life screening

campaigns. However, this possibility is never discussed, neither theoretically, nor in the context of the obtained results. Noteworthy, the authors never screened any randomly mutagenized libraries – here many false-positive hits can be expected with these biosensors.

In the discussion, this idea of having developed BCHA-specific biosensors (for the first time!!!) is picked up again when the authors state that “no genetically encoded biosensor for this class of chemicals has been reported in this organism” (line 566). The authors are clearer in the next sentences, but I still think that this “sales-strategy” is not right.

For the characterization of the isobutanol biosensor, the authors supplemented two different metabolites to determine the biosensor response (line 161). Well, what about ligand uptake? Are they equally well taken up? α -IPM and α -KIV are chemically quite different. Maybe it would have been useful to test a possible response to other chemically not so different compounds such as α -keto-3-methylvalerate. This compound might not be relevant for the two desired products, but could help to explain occurrence of possible false-positives. Validation is also important since the authors claim an “exclusive response” in the discussion (l. 572). Furthermore, I would have loved to see the point at which the linear response is lost (for α -IPM as well as α -KIV).

Protein degradation tags have a strong influence on the protein stability (as desired by the authors) but also have a negative impact on the quantum yield (and sometimes on the brightness). This should be discussed. The reason for fusing the PEST-tag is a bit confusing. The saturation of the biosensor with excess α -IPM occurs at the promoter of the biosensor. Hence, addition of a degradation tag to the reporter protein would only make the biosensor (more) dynamic. A direct comparison of the two sensor variants in the same isobutanol strain background might shed some light on this.

Another concern is that there is a lot of guessing when it comes to “metabolic flux” in a pathway leading to byproduct/intermediate accumulation, which dictated the biosensor design. The intracellular concentrations should be determined.

The authors used the “isobutanol biosensor” to isolate isobutanol-producing strains. What is the library size? “Two rounds of FACS” are mentioned (l. 198), but the term “enrichment” is missing. It is not really a surprise that the average product titer increased and that more improved producers can be found after the second round of FACS. However, the reader is left with the question: Is it several times the same variant (or only a few different ones)? Probably yes – knowledge of the library size would have given the reader a feeling, sequencing would have provided certainty. The question of the library size does also apply to the other sections in which targeted-mutagenesis libraries were screened. In the end this is also true for the cytosolic pathway engineering where the random insertion of a cassette appears to be the only source of genetic diversity during the initial engineering step.

In the discussion (line 646) the wording is again misleading when the authors claim to have developed the “first mitochondrial biosensor based on a transcription factor”. The biosensor is definitely not mitochondrial biosensors as it is encoded and located in the cytoplasm.

Minor comments:

line 111 – 117 is a repetition – the same information can be found in the introduction (see comment regarding the length of the manuscript)

line 134 - I would assume that a LEU2 deletion strain is leucine-auxotroph. That might not be relevant during screening for isobutanol, but should at least be mentioned

line 169 - the “applicability of this biosensor over at least an 11-fold range of isobutanol production” sounds very good, but we are talking about the low mg/L/OD-range, so this does not mean much. (This is also true for the “other” biosensor (line 187))

line 581 – The authors claim that monitoring of an intracellular metabolite as opposed to the end product is better as the desired alcohols cross membranes. I agree that cross talk might pose a problem. However, this can be easily fixed by working with diluted cultures during FACS as many colleagues have shown. The isolation of false-positives poses a much larger challenge (see above). This is in particular true when (really) randomly mutated (chemical mutagenesis, UV, etc.), larger libraries need to be analyzed.

line 633 – no, in an industrial setting biosensors for valine and leucine would be (and are) used for the reasons outlined here.
Supplementary figure 1, it appears as if the conversion of glucose to pyruvate (and ethanol) happens outside of the yeast cell.

Reviewer #3:

Remarks to the Author:

The title of the paper suggests a sensor development. However, only the first section out of the in total 5 result sections of the manuscript covers the sensor development. The remaining four sections are only different (and a bit repetitive) application cases of the sensor. Thus, I didn't get the content I was expecting to get after having read the title. In fact, I was wondering what the main contribution of the paper would be. Is it (i) the sensor development, or (ii) some yeast strains with improved isobutanol or isopentanol productivity? The title suggests that it is (i), the introduction is more geared towards (ii), the core of the manuscript for 1/5th covers (i) and for 4/5th covers (ii). If the authors stick to (i) as being the main aim, then I would expect more comprehensive work on characterizing the sensor (i.e. calibration curves, etc.). If the main is (ii), then "some" sensor that in an empiric manner reports on productivity is fine, but then we judge the work on the actual strain improvements. Overall, I feel there are different options for how the manuscript could be positioned. I feel that at the moment the manuscript is somewhere hanging in the middle.

Right now, I am evaluating the manuscript according to its main aim as stated in the title, i.e. the development of a sensor. I am not sure whether I precisely understood how the sensor really functions. The transcription factor binds to a metabolic intermediate. Thus, primarily the sensor should be a sensor for the concentration of this intermediate. However, throughout the text the authors also imply that the sensor reports on the flux through the pathway. On the practical level the sensor output also seems to correlate with the productivity (i.e. a rate) of the strains, but I cannot get my head around this. I feel something fundamental is missing here (or I might not have gotten it). Also, gene/protein expression and (thus also the GFP expression driven by the sensor) should be a function of the cellular growth rate. How does this factor in here? Would cells with different growth rate but otherwise same alpha-IPM concentration have the same sensor output? Also, I am a bit concerned about the fact that the sensor output (fluorescence) was measured during exponential phase (13 h), while the product titers were measured only after 48 h (which is surely not exponential anymore): What if all product was only produced in the late exponential/early stationary phase? Thus, I am wondering how does fluorescence in the exponential phase mechanistically correlate with the final product titers? Is the product synthesis rate constant in the whole cultivation period, i.e. until 48h? While the practical application cases shown by the authors show that the "sensor" indeed allows to pick up mutants with increased product titers, I feel that for publication in NComm, we need more mechanistic understanding on how the sensor works, and what it is actually reporting (an intermediate concentration, a flux) and why.

Response to reviewers' comments

We thank the reviewers for carefully reading our manuscript and providing valuable feedback, which helped us improve its quality and strengthen our conclusions. In our effort to address every concern raised by the reviewers and conduct all the experiments they requested, we recruited new authors with expertise in different techniques. However, the departure of the first and second authors of the original manuscript to new positions outside the University, compounded with personal difficulties faced by several authors due to the current pandemic, caused an unusually long delay in the submission of this revised manuscript. We are very grateful to *Nature Communications* for their flexibility and patience in accepting this revised version, and equally grateful to the reviewers for their willingness to review it after this unusually long revision period. Below, we provide point-by-point responses to the reviewers' comments and descriptions of the changes we have made to the manuscript to address their concerns.

REVIEWER COMMENTS

Reviewer #1 (Remarks to the Author):

The manuscript of Zhang et al. describes an impressive and interesting work in industrial biotechnology with yeast. The authors developed a biosensor for an intermediate of branched-chain amino acid metabolism (IPM) and used the biosensor in two different set-ups to perform high-throughput screens for improved isobutanol and isopentanol production. The biosensor turned out to be quite effective for this. It will be extremely valuable to further improve isobutanol and isopentanol production with yeast, two very promising biofuels.

I have only some minor comments:

We thank the reviewer for their supportive comments.

1) The authors claim that they have developed two biosensors. However, it is rather only one biosensor which is used in two different set-ups.

The reviewer is correct in that the core of the isobutanol and isopentanol biosensors consists of the same α -IPM-mediated Leu3p control of the *LEU1* promoter (P_{LEU1}) driving yEGFP expression. However, there are three significant differences in the design of each biosensor (the Leu4p variant, presence or absence of a PEST tag fused to C-terminus of yEGFP, and the presence or absence of *LEU2*). These different designs display linear correlations between GFP fluorescence and either isobutanol or isopentanol production, but not both (Supplementary Figure 5). Therefore, to address the reviewer's comment while also avoiding confusion between the two different designs (or set-ups), we have changed the text to reflect that we have developed one biosensor with two configurations. Instead of referring to two biosensors, we refer to the first design as the biosensor in the isobutanol configuration (isobutanol-configured biosensor) and the second design as the biosensor in the isopentanol configuration (isopentanol-configured biosensor).

2) L. 134: it should be explained why *LEU2* was deleted and not *LEU1* which actually encodes the enzyme converting IPM. Otherwise it is confusing.

This is a good question. Our isobutanol-producing strains take advantage of the *LEU2* auxotrophic marker present in the CEN.PK2-1C parent strain (MATa *ura3-52 trp1-289 leu2-3,112 his3-1 MAL2-8c SUC2*). With this *LEU2* deletion present, it is not necessary to also delete *LEU1* to prevent losing flux towards leucine biosynthesis. Furthermore, we hypothesize that the reversible activity of Leu1p could help reduce the biosensor background by dampening the accumulation of α -IPM. We have added these explanations to the revised manuscript, specifically towards the end of the first paragraph in the **new Supplementary Note 1**, which now contains the details of the biosensor design to reduce the length of the main text.

3) The authors several times stress that it is very important that their new biosensor can be used to optimize as well a mitochondrial as also a cytosolic isobutanol pathway. Why? This is obvious. The biosensor actually is completely independent from intracellular localization of the pathway, especially as also the intermediates of branched-chain amino acid metabolism seem to be permeable to membranes. Therefore, this statement seems to be unnecessary and confusing.

The reviewer's concern is understandable, and we appreciate that they find it obvious. However, the reason why we stress this feature is not because it is an unexpected finding, but because two major strategies have been used to engineer *Saccharomyces cerevisiae* for isobutanol production: by fully compartmentalizing the biosynthetic pathway in mitochondria or cytosol. Therefore, it is an advantage that this biosensor can be used to improve isobutanol-producing strains developed through either metabolic engineering approach (we mention this point in Lines 335-337 and Lines 482-485). Nevertheless, we concede that there is no need to overemphasize this point, especially for the less-specialized reader. Therefore, we have de-emphasized this point where appropriate, including removing it from the title of the manuscript.

4) The authors do not discuss the limits of their new biosensor. As it detects an intermediate in metabolism it is only useful to screen for improvements in the upstream part of the pathway. If the downstream Ehrlich pathway would be or would become limiting the sensor could not be used any longer. Or can it be used then to detect a decrease in IPM concentrations? This should be discussed.

The reviewer brings up a good point, and one we wish to clarify. However, we should note that it is not uncommon for biosensors to detect a pathway intermediate, and has been shown in several biosensors engineered for other products (Zhang, F.Z., Carothers, J.M. & Keasling, J.D. *Nature Biotechnology* **30**, 354-U166 (2012). DOI: [10.1038/nbt.2149](https://doi.org/10.1038/nbt.2149); Rogers J.K. & Church, G.M. *PNAS* **113** (9) 2388-2393 (2016); <https://doi.org/10.1073/pnas.1600375113>).

We have now added new discussion on this point in Lines 502-509:

“This high efficiency is likely due to the fact that the source of diversity in all our applications is upstream of the α -IPM intermediate, for which the biosensor was designed. While most of the biosynthetic pathway and previously identified bottlenecks lay upstream of α -IPM, if the downstream pathway became limiting, the rate of false positives for isobutanol production could increase. Similarly, we do not expect the same high-throughput screens used in this study to be effective at identifying enhanced enzymes downstream of α -IPM. However, sorting strains with decreased fluorescence, reflecting increased rates of α -IPM processing, might be helpful in these scenarios.”

5) L. 581-585: the conclusion is not correct. The authors showed on page 8 (line 164-165) that extracellular addition of KIV and IPM stimulated the biosensor, demonstrating permeability. Therefore, also these intermediate are permeable and could traverse membranes.

We thank the reviewer for catching this apparent contradiction. While it is true that we show that extracellular α -KIV and α -IPM can be imported into the cell, there is no evidence that α -KIV or α -IPM are secreted from cells into the medium at any level that would activate our biosensors, and certainly far from the levels of final product (e.g., isobutanol or isopentanol) that would be expected to accumulate. Therefore, our statement in the conclusion is technically correct. Nevertheless, to avoid confusion, we have reworded the statement to say: "...monitoring the concentration of an intracellular metabolite (\square -IPM), as opposed to the concentration of a secreted end product that easily traverses cell membranes bi-directionally...", to make this distinction clearer and more accurate (Lines 478-479).

6) 1500% and 2000% should be replaced with 15-fold and 20-fold.

We have changed 1500% and 2000% to 15-fold and 20-fold, respectively (Line 495).

Reviewer #2 (Remarks to the Author):

In this manuscript, Zhang and coworkers present the design and construction two biosensors for the indirect detection of isobutanol and isopentanol in yeast. Following, several examples are presented in which the biosensors were used in combination with FACS to isolate isobutanol producers or to engineer enzymes (relieve from feedback-inhibition; increased enzymatic activity) involved in BCAA-biosynthesis. Finally, a mitochondrial- and a cytosolic pathway for isobutanol are presented.

The manuscript is well written, but way too long. In particular, the introduction and the discussion could be substantially shorter. In addition, one or even two application examples could removed from the text, e.g. the isolation of isobutanol-producing strains as this is a simple enrichment.

Unfortunately, the abstract promises much more than the manuscript can keep: The Leu3p-based transcriptional biosensor does not detect any of the desired products but only a byproduct or intermediate (depending on the product, see major remarks below). In addition, the biosensor characterization is not very detailed as the operational range is only partially investigated the dynamic range is not discussed at all. In addition, some important experiments such as intracellular quantification of α -IPM in the different biosensor-setsups are also missing (see below).

Taken together, I think that this manuscript is more appropriate for a journal such as ACS Synthetic Biology.

We thank the reviewer for their feedback. We have significantly shortened the length of the manuscript, especially the introduction and discussion sections. To achieve this, we simplified the text, removed redundant information, and moved some of the details and tangential information to **seven new Supplementary Notes**. We have also clarified in the abstract that the biosensor acts by detecting intracellular levels of α -IPM. It should be noted, however, that it is not uncommon for

biosensors to work by detecting metabolic precursors or byproducts (see response below), and that these types of biosensors offer unique advantages, such as more accurate detection of cellular level production as opposed to the average production at the fermentation scale (see Discussion section). In addition, we include a more detailed discussion on the operational and dynamic range of the biosensor, (**new Supplementary Note 2**). **We also conducted new experiments to quantify the intracellular concentrations of α -IPM in different biosensor setups**, as the reviewer requested, (**new Supplementary Figure 4**), which are discussed in detail in a **new Supplementary Note 3**. Additional details to comments raised in this paragraph are specifically addressed below.

Major remarks:

1) In general, I find the term “isobutanol biosensor” (e.g. line 128) or “isopentanol biosensor” very misleading as either a byproduct or a pathway intermediate is detected. In addition, except for the PEST-tag the sensors are essentially the same. It is the pathway configuration, which does the trick.

It is certainly not our intention to mislead. Biosensors are commonly named after the final product for which they are designed, even if their molecular mechanism is based on sensing a precursor metabolite. For example, a biosensor was previously described (and named) for production of fatty-acid-derived chemicals even though it detects acyl-CoAs (Zhang, F.Z., Carothers, J.M. & Keasling, J.D. *Nature Biotechnology* **30**, (4) 354-9 U166 (2012). DOI: 10.1038/nbt.2149). In another example, 3-hydroxypropionate biosensors detect the precursor 2-methylcitrate or byproduct acrylate (Rogers J.K. & Church, G.M. *PNAS* **113** (9) 2388-2393 (2016); <https://doi.org/10.1073/pnas.1600375113>). This accepted practice is meant to use a name that conveys the application of the biosensor, which is more informative, rather than using a name reflective of its mechanistic details. In our description of the mechanism, and throughout the manuscript, we clearly explain that our biosensors detect α -IPM, and not alcohols, so there is no misrepresentation of our biosensors. In fact, we consider this to be an advantage of our biosensors, as this could potentially allow them to be used in autoregulatory systems or for other BCAA-derived products, as stated in the Discussion.

The reviewer is correct in that the core of both the isobutanol and isopentanol biosensors consists of α -IPM-mediated Leu3p control of the *LEU1* promoter (P_{LEU1}) driving yEGFP. However, as clarified for Reviewer 1, there are *three* significant differences in the design of each biosensor: 1) the Leu4p variant; 2) the presence or absence of a PEST tag fused to the yEGFP reporter, and 3) the presence or absence of *LEU2*. Additionally, we show that these different set-ups measure either isobutanol or isopentanol production, but not both (Supplementary Figure 5). Therefore, it would be confusing to the reader if we only wrote about one biosensor. Nevertheless, we concede that referring to these different designs as “configurations” is a good solution to avoid confusion while addressing the reviewer’s concern. We have changed the text to reflect that we have developed *one* biosensor with *two configurations*. Instead of referring to two biosensors, we refer to the first design as the biosensor in the isobutanol configuration (isobutanol-configured biosensor) and the second design as the biosensor in the isopentanol configuration (isopentanol-configured biosensor).

2) In addition, it is great to see that engineered strains give a good biosensor response, but when only a byproduct/intermediate and not the desired product is detected, a large number of false

positive variants (no increased product titer) can be expected in a real-life screening campaigns. However, this possibility is never discussed, neither theoretically, nor in the context of the obtained results. Noteworthy, the authors never screened any randomly mutagenized libraries – here many false-positive hits can be expected with these biosensors.

We are glad the reviewer agrees that our biosensor has a good response. We are not sure what the reviewer means by “real-life screening campaigns”. However, the rates of false positives were too low to be detected in any of the five different screens we performed with our biosensors. Nevertheless, as the reviewer argues, it is theoretically possible that under some other conditions or other untested screens the rates of false positives could be higher. To discuss this important aspect of the study, we have included a new paragraph in the discussion (Lines 497-511), in which we specifically disclose the potential of higher false positives in certain scenarios. Specifically, we wrote:

“While most of the biosynthetic pathway and previously identified bottlenecks lay upstream of α -IPM, if the downstream pathway became limiting, the rate of false positives for isobutanol production could increase.” (Lines 504-506).

We are not sure what exactly the reviewer means by saying that we never screened randomly mutagenized libraries. In fact, we screened randomly mutagenized libraries of *ILV6*, *LEU4*, and *Ll_ilvD* and found no false positives after two or three rounds of sorting. This is illustrated by comparing isobutanol or isopentanol production of 120 randomly selected, sorted colonies to production of controls containing wild type version of the enzymes in Supplementary Figures 6, 9, and 13a. To further illustrate the low rate of false negatives obtained from screens of randomly mutagenized enzymes (*ILV6*, *LEU4* and *ilvD*), we have added a **new Supplementary Figure 15**. In this figure, we graph the fluorescence signal measured for all enzyme variants tested against the actual isobutanol or isopentanol produced by each variant. This confirms that the correlation between biosensor output and production remains high in randomly generated libraries.

Perhaps with that comment the reviewer meant to say a library of random mutations in the genome, which we could have certainly done, but we believe that we have provided enough examples of applications to demonstrate the utility of our two biosensor configurations, as well as a sense for the low rates of false positives obtained thus far. This sentiment is probably shared by this reviewer, based on the above recommendation to remove one or two application examples. Furthermore, genomic mutagenesis screens, which we are actively pursuing with promising preliminary results, are likely to yield new insights that warrant deeper investigation, and is thus beyond the scope of this study. Nevertheless, to disclose the possibility of obtaining higher false negatives in genomic screens, we now add the following text: “Furthermore, screening for random mutations in the genome could also lead to higher rates of false positives.” (Lines 509-510).

3) In the discussion, this idea of having developed BCHA-specific biosensors (for the first time!!!) is picked up again when the authors state that “no genetically encoded biosensor for this class of chemicals has been reported in this organism” (line 566). The authors are clearer in the next sentences, but I still think that this “sales-strategy” is not right.

We do not understand why the reviewer seems to imply that we use a disingenuous “sales-strategy” that “is not right” to present our new biosensor. We are clear and transparent in the Discussion about the context in which it is placed in the field. In the original manuscript, as well as the revised version, we cite the previous biosensor developed for isobutanol in *E. coli* based on the transcription factor BmoR (Ref. 60), which is not specific for BCHAs, as this transcription factor cross-reacts with C4-C6 linear alcohols, C3-C5 branched alcohols, and even dicarboxylic acids (Ref. 61). Moreover, being a prokaryotic transcription factor, BmoR biosensors have not been applied in yeast. We also cite a biosensor developed in yeast for 1-butanol (Ref. 62), which, while being a higher alcohol, is not derived from branched chain amino acids and therefore is not a BCHA; furthermore, this biosensor can cross-react with other cellular stresses. Finally, we also cite a FRET biosensor previously developed for branched chain amino acids in mammalian cells (Refs. 67, 68), which can only measure the collective intracellular concentrations of valine, leucine, and isoleucine, and has not been demonstrated in yeast or for the alcohols derived from them (BCHAs).

In contrast to all these previous biosensors, our biosensor uses Leu3p, which specifically senses α -IPM, an intermediate of BCHA biosynthesis, thus allowing us to implement two biosensor designs that specifically report isobutanol or isopentanol production (Figure 1 and Supplementary Figure 5). Therefore, we stand by our assertion that no genetically encoded biosensor specific to this class of chemicals (BCHAs) has been previously reported in yeast, which is a standard way to explain why a new technology cannot be easily compared with previous developments. However, if the reviewer knows of any previously reported biosensors specific for BCHA production in yeast we will be happy to cite them in our study and compare them to the biosensor we have developed.

Nevertheless, to address the reviewer’s concern we no longer use the term “first” to describe our biosensor.

4) For the characterization of the isobutanol biosensor, the authors supplemented two different metabolites to determine the biosensor response (line 161). Well, what about ligand uptake? Are they equally well taken up? α -IPM and α -KIV are chemically quite different. Maybe it would have been useful to test a possible response to other chemically not so different compounds such as α -keto-3-methylvalerate. This compound might not be relevant for the two desired products, but could help to explain occurrence of possible false-positives. Validation is also important since the authors claim an “exclusive response” in the discussion (l. 572). Furthermore, I would have loved to see the point at which the linear response is lost (for α -IPM as well as α -KIV).

We thank the reviewer for their suggestions. We have added **new data to show that the isobutanol biosensor is unresponsive to α -keto-3-methylvalerate (α -K3MV, new Supplementary Figures 3e,f)** supplemented in the media. To address the reviewer’s concern, we have also added the following sentence to the main text (Lines 145-147), explaining the new results: “In contrast, the biosensor does not respond to the isoleucine precursor α -K3MV, when supplemented in the media (**Supplementary Fig. 3e,f**).”

Also, as suggested by the reviewer, we tested the response of the biosensor at higher concentrations of α -IPM and α -KIV (**new Supplementary Figures 3c,d**). We found that the biosensor response

saturates when feeding α -IPM above $\sim 80 \mu\text{M}$, or α -KIV above $\sim 300 \mu\text{M}$. Since the stoichiometric ratio between these metabolites in the biosynthetic pathway is one to one, and the biosensor response is linear with respect to both metabolites up to $\sim 80 \mu\text{M}$, the saturation at different concentrations suggests there may be differences in the rates of cellular uptake of these metabolites, as the reviewer predicted, which likely become limiting at different concentrations. Similarly, differences in their metabolic conversion rates could greatly affect the biosensor response when feeding metabolites to the media. Therefore, caution should be taken when interpreting the results from these feeding experiments, which are unlikely to reflect the true dynamic range of the biosensor when these metabolites are instead produced inside the cell in the BCAA biosynthetic pathway. Rather, these experiments were conducted to confirm that the biosensor could respond to elevated levels of α -IPM and α -KIV, as expected. Measuring the biosensor response using strains engineered to produce different levels of isobutanol or isopentanol (and thus produce different levels of α -IPM and α -KIV) is a more direct approach to studying the performance of the biosensor as it is used in real applications; however, this method is still limited by the maximal productivity achieved in our best engineered strains (Figure 1b,d). Therefore, these experiments provide at best a conservative estimate of the true dynamic range of the biosensor. These limitations are discussed in a **new Supplementary Note 2**.

5) Protein degradation tags have a strong influence on the protein stability (as desired by the authors) but also have a negative impact on the quantum yield (and sometimes on the brightness). This should be discussed. The reason for fusing the PEST-tag is a bit confusing. The saturation of the biosensor with excess α -IPM occurs at the promoter of the biosensor. Hence, addition of a degradation tag to the reporter protein would only make the biosensor (more) dynamic. A direct comparison of the two sensor variants in the same isobutanol strain background might shed some light on this.

As the reviewer suggested, **we now show new experimental data in which we compare the performance of different biosensor configurations using a GFP reporter with or without a PEST-tag in the same strain background (new Supplementary Figure 2)**. These new data are discussed in a **new Supplementary Note 1**, and further supports our biosensor designs. The reason why it is necessary to add a PEST-tag to the reporter in the isobutanol biosensor configuration but not for the isopentanol configuration is that there is a large accumulation of α -IPM in the former that does not occur in the latter. This is caused by the *LEU2* deletion in the isobutanol biosensor configuration, which leads to α -IPM accumulation; whereas in the isopentanol configuration, *LEU2* is present and its activity significantly reduces intracellular α -IPM concentrations as it is channeled towards isopentanol production (Figure 1 and Supplementary Figure 1). This is supported by **new experiments we conducted to measure intracellular concentrations of α -IPM (new Supplementary Figure 4b, e)**, which indeed show that the concentration of α -IPM is more than 5-fold higher in the isobutanol configuration than in the isopentanol configuration. Therefore, because Leu3p activity responds to intracellular α -IPM concentrations, expression of the GFP reporter is significantly higher in the isobutanol-configured biosensor than in the isopentanol-configured biosensor. Consequently, the PEST-tag is required in the isobutanol configuration to prevent high background and possibly early saturation of the biosensor readout, whereas adding a PEST-tag to the isopentanol configuration reduces the sensitivity of the biosensor to levels below the limit of detection. These mechanistic insights are now discussed in a **new Supplementary Note 3**.

Regarding the possibility of the PEST-tag affecting the quantum yield or brightness of GFP, we found several studies showing that this does not occur with small protein fusions, including the PEST tag from Cln2 used in this study (Mateus C. & Avery S.V., *Yeast* **16** (14), 1313-1323 (2000), DOI:[https://doi.org/10.1002/1097-0061\(200010\)16:14<1313::AID-YEA626>3.0.CO;2-O](https://doi.org/10.1002/1097-0061(200010)16:14<1313::AID-YEA626>3.0.CO;2-O); Weber-Ban E.U., et.al., *Nature* **401**, 90-93 (1999) DOI: <https://www.nature.com/articles/43481>; Defenbaugh, D.A. & Nakai, H., *JBC* **278** (5) 52333-52339 (2003), DOI:<https://doi.org/10.1074/jbc.M308724200>; Martinez V., et.al., *Nucleic Acids Research* **45** (21) e171 (2017), DOI: <https://doi.org/10.1093/nar/gkx797>), which we now cite. However, we cannot rule out that the PEST-tag may be causing some effect on GFP fluorescence. Therefore, we have added the following sentence in the **new Supplementary Note 1**: “It is important to note that previous studies have shown that small protein fusions cause no measurable effect on GFP quantum yield or brightness, including the PEST-Cln2 tag used in this study, which does not seem to impede biosensor functionality.”

6) Another concern is that there is a lot of guessing when it comes to “metabolic flux” in a pathway leading to byproduct/intermediate accumulation, which dictated the biosensor design. The intracellular concentrations should be determined.

To address the reviewers’ concern, we measured intracellular concentrations of α -IPM and α -KIV in isobutanol- and isopentanol-producing strains using UHPLC-Orbitrap mass spectrometry (**new Supplementary Figure 4**). Consistent with our original hypothesis, we found that the intracellular α -IPM concentrations are higher in BCHA-producing strains than in negative controls, and that this concentration is correlated with the biosensors’ fluorescence as well as BCHA production. In all cases, the intracellular levels of α -KIV were below the limit of detection, suggesting that this metabolite is rapidly consumed (likely by Leu4p, in addition to decarboxylases and transaminases). Furthermore, our original hypothesis that blocking metabolic flux towards leucine biosynthesis (via *LEU2* deletion) would lead to intracellular α -IPM accumulation is now confirmed by our **new measurements of intracellular levels of this metabolite**. The mechanistic insights derived from these new experiments are discussed in a **new Supplementary Note 3**. Additionally, to address the reviewers’ concern regarding the use of the term “metabolic flux”, we have revised it to say “metabolic activity” where appropriate (Lines 75, 530)

7) The authors used the “isobutanol biosensor” to isolate isobutanol-producing strains. What is the library size? “Two rounds of FACS” are mentioned (l. 198), but the term “enrichment” is missing. It is not really a surprise that the average product titer increased and that more improved producers can be found after the second round of FACS. However, the reader is left with the question: Is it several times the same variant (or only a few different ones)? Probably yes – knowledge of the library size would have given the reader a feeling, sequencing would have provided certainty. The question of the library size does also apply to the other sections in which targeted-mutagenesis libraries were screened. In the end this is also true for the cytosolic pathway engineering where the random insertion of a cassette appears to be the only source of genetic diversity during the initial engineering step.

Unfortunately, the unsorted library of combinatorial strains (Figure 2) is not barcoded, which makes it difficult to precisely measure library size and enrichment. However, to address the reviewer's concern, we estimated the maximum size of the original library based on the efficiency of library transformation and the number of calculated transformants derived from colony counts of serial dilutions. With this method, we estimate a maximum library size of $\sim 2 \times 10^5$, which we now include in Lines 183 and 653. Additionally, **we conducted new experiments using digital droplet PCR to obtain an average genotype of the pre-sorted library and individual genotypes of the top six isobutanol-producing strains obtained after the second FACS round (new Supplementary Table 3)**. Of the top six producing strains we found only two distinct genotypes, demonstrating that we obtain only a “few variants” after two rounds of FACS, as the reviewer correctly predicted. While we are unable to make a precise measurement of enrichment, this new data gives a good estimate, based on the maximum size and average diversity of the pre-sorted library and the reduced diversity of the sorted strains. We now briefly discuss these findings in Lines 181-184: "using digital droplet PCR (see methods), we found only two different genotypes in the top 6 isobutanol-producing strains (those producing 700 mg/L isobutanol or more), suggesting a substantial enrichment from the $\sim 2 \times 10^5$ individual transformants in the original library”.

Similarly, we were unable to precisely measure the library sizes and enrichment in FACS experiments done on PCR mutagenesis libraries of metabolic enzymes. However, based on the efficiency of our mutagenesis method, and the number of yeast colonies obtained in our transformations, we estimate pre-sorted library sizes of 10^6 - 10^7 , which is now described in Lines 624-631. Moreover, sequencing randomly picked variants from colonies obtained after two rounds of FACS for each unbiased mutagenesis experiment revealed that key mutations were overrepresented, suggesting a substantial enrichment of several orders of magnitude for mutations that enhance enzymatic activity. For example, nine out of ten randomly picked sorted mutants obtained from mutagenizing wild type *ILV6* contain one or more mutations in residues located in the putative valine-binding site or its vicinity (N86, V90, L91, N104, V110, and/or E133; See Supplementary Table 4 and Supplementary Figure 8). Additionally, nine out of 13 randomly picked sorted mutants obtained from mutagenizing the full-length *LEU4* contain mutations in residues located in the leucine binding site or its vicinity (Y485, F497, N515, Y538, H541, D578, V584, T590; See Supplementary Tables 5 and 6, and Supplementary Figure 11). Finally, the mutation that most significantly enhances Ll_ilvD, I433V, was isolated in three independent mutagenesis plus sorting (three FACS rounds) experiments.

Therefore, while we are unable to provide a precise measurement of library enrichment for these experiments, our biosensors repeatedly showed their ability to isolate significantly enhanced strains or enzyme variants from pre-sorted libraries that varied in size from $\sim 10^5 - 10^7$ and had average background activities indistinguishable from wild type.

8) In the discussion (line 646) the wording is again misleading when the authors claim to have developed the “first mitochondrial biosensor based on a transcription factor”. The biosensor is definitely not mitochondrial biosensors as it is encoded and located in the cytoplasm.

Again, our intention was not to mislead, and we regret that the wording may have caused some confusion. We have edited the text to clarify that our biosensor monitors a mitochondrial metabolic

pathway (avoiding any misinterpretation that Leu3p is a mitochondrial transcription factor), as follows: “While previously reported biosensors based on GFP-ligand-binding-protein chimeras, FRET, or its bioluminescence equivalent (BRET) have been developed to probe the mitochondrial metabolic state⁶⁶, our biosensor is capable of probing the activity of a mitochondrial biosynthetic pathway.” (Line 528-531).

Minor comments:

9) line 111 – 117 is a repetition – the same information can be found in the introduction (see comment regarding the length of the manuscript)

We agree, and have reduced and edited the text to avoid repetition, as recommended by the reviewer.

10) line 134 - I would assume that a LEU2 deletion strain is leucine-auxotroph. That might not be relevant during screening for isobutanol, but should at least be mentioned

We agree and thank the reviewer for the suggestion. We now include the text: “*LEU2* deletion also causes leucine auxotrophy, requiring supplementation of this amino acid in the growth medium” in **new Supplementary Note 1**, which provides additional details on the development and evaluation of different biosensor constructs.

11) line 169 - the “applicability of this biosensor over at least an 11-fold range of isobutanol production” sounds very good, but we are talking about the low mg/L/OD-range, so this does not mean much. (This is also true for the “other” biosensor (line 187))

We agree that the dynamic ranges of our two biosensor configurations were calculated over a relatively low mg/L/OD range and thus we no longer report the 11-fold change. However, it is important to note that we are unable to fully explore the true limits and dynamic range of the biosensor in its different configurations as it is difficult to precisely control the levels of intracellular α -IPM in our experiments. Feeding α -IPM or α -KIV in the media presents limits due to uncertainties in the cell’s uptake and metabolic processing rates, while using engineered BCHA producing strains is limited by the maximal productivity of our most productive engineered strains, which is still relatively low. Therefore the dynamic ranges obtained through these methods should be taken as conservative estimates of their true values and serve mostly to confirm the response of the biosensor to increasing levels of α -IPM (see our response to Question 4 from this Reviewer above). We added a **new Supplementary Note 2** to address this concern and bring attention to the limits of our experiments.

12) line 581 – The authors claim that monitoring of an intracellular metabolite as opposed to the end product is better as the desired alcohols cross membranes. I agree that cross talk might pose a problem. However, this can be easily fixed by working with diluted cultures during FACS as many colleagues have shown. The isolation of false-positives poses a much larger challenge (see above). This is in particular true when (really) randomly mutated (chemical mutagenesis, UV, etc.), larger libraries need to be analyzed.

We thank the reviewer for the valuable input. We have removed the word “better” from this line in the text, and edited it to say that our “... biosensors facilitate the detection of BCAA-derived product biosynthesis in each individual cell...” (Line 480). While we agree that “false-positives” can pose challenges in the applications of biosensors, we did not find a measurable number of false-positives in any of our screens (see our response to Question 2 of this Reviewer above).

line 633 – no, in an industrial setting biosensors for valine and leucine would be (and are) used for the reasons outlined here.

We were unable to find any valine or leucine biosensors in yeast that are being used in laboratory or industrial settings. We would be very grateful if the reviewer provides a reference so we can cite them.

Supplementary figure 1, it appears as if the conversion of glucose to pyruvate (and ethanol) happens outside of the yeast cell.

We thank the reviewer for bringing this to our attention. We have added a new compartment in Supplementary Figure 1 to delineate all cytosolic components, including the glycolytic pathway and ethanol fermentation. We also now write the word “Cytosol” closer to these pathways to avoid confusion.

Reviewer #3 (Remarks to the Author):

The title of the paper suggests a sensor development. However, only the first section out of the in total 5 result sections of the manuscript covers the sensor development. The remaining four sections are only different (and a bit repetitive) application cases of the sensor. Thus, I didn't get the content I was expecting to get after having read the title. In fact, I was wondering what the main contribution of the paper would be. Is it (i) the sensor development, or (ii) some yeast strains with improved isobutanol or isopentanol productivity? The title suggests that it is (i), the introduction is more geared towards (ii), the core of the manuscript for 1/5th covers (i) and for 4/5th covers (ii). If the authors stick to (i) as being the main aim, then I would expect more comprehensive work on characterizing the sensor (i.e. calibration curves, etc.). If the main is (ii), then “some” sensor that in an empiric manner reports on productivity is fine, but then we judge the work on the actual strain improvements. Overall, I feel there are different options for how the manuscript could be positioned. I feel that at the moment the manuscript is somewhere hanging in the middle.

We thank the reviewer for the insightful comments. Our manuscript intends to address both the development and applications of the biosensors. Many previously reported biosensors have been characterized, but only minimally demonstrated. Thus, we sought to both develop the biosensors, and then demonstrate different applications such that it is clear how they could be applied to aid in high-throughput screening for strain development and enzyme engineering. We regret that the different demonstrations seemed repetitive and we have edited the text to reduce repetition; however we believe that each one serves a unique purpose: **1)** The section the reviewer calls 2/5 (Figure 2) demonstrates that the isobutanol-configured biosensor can be used to isolate

productive strains from a library of combinatorially assembled upstream and downstream pathways (a task that is currently performed through laborious screening of individual strains); **2)** In section 3/5 (Figure 3), we show that the isobutanol-configured biosensor can be used to engineer endogenous mitochondrial enzymes (*ILV6*); **3)** Section 4/5 (Figure 4), contains the only demonstration of the isopentanol-configured biosensor, and features its ability to enhance an endogenous enzyme, Leu4p, that exists in both mitochondria and cytosol; **4)** Section 5/5 (Figure 5) shows that the isobutanol-configured biosensor can also be used to improve heterologous enzymes in the cytosolic compartment. We chose to focus on *ilvD* because its use of Fe/S clusters makes it arguably the most challenging enzyme in the pathway to optimize (this is reflected by the fact that no hyperactive mutants had been previously reported). Therefore, the apparent disproportionate number of applications in our study comes from the fact that we aimed to demonstrate applications of both the isobutanol and isopentanol configurations of the biosensor, as well as demonstrate that the biosensor works in conjunction with the two different approaches the field has taken to produce BCHAs in yeast: complete mitochondrial or cytosolic metabolic compartmentalization. Since each demonstration adds unique value to the capabilities of our biosensor, we believe it is important to keep them all.

Nevertheless, to improve the balance between biosensor development and demonstration, we now include additional information on biosensor design and characterization. **We provide new descriptions of different designs we tested during development of the isobutanol and isopentanol biosensor configurations (new Supplementary Note 1) and new characterization data (new Supplementary Figure 2) showing their performance when using different insensitive Leu4p variants, different GFP reporter stabilities (by optional use of a PEST-tag), and either a *LEU2* or *leu2Δ* genetic background (which ultimately defined the isobutanol and isopentanol configurations).** The new data (**new Supplementary Figure 2**) shows the effects that different combinations of these features have on biosensor sensitivity, background signal, and apparent range using BCHA-producing strains. These results enable us to more clearly justify our choice of biosensor designs. Additionally, **we provide new biosensor response data when supplementing the media with higher concentrations of α -KIV and α -IPM (new Supplementary Figure 3c,d),** which suggest this approach to characterizing the biosensor may be limited by cellular importation and downstream processing of these metabolites. We also provide **new data showing the high correlations obtained between biosensor signals (in both the isobutanol and isopentanol configurations) and BCHA production in strains obtained from three different biosensor applications,** which show the low rates of false positives sorted with our biosensors (**new Supplementary Figure 15**). Furthermore, we provide **new data to correlate intracellular α -IPM concentrations with biosensor response and BCHA production (new Supplementary Figure 4),** which we describe in more detail in the response below.

Finally, we modified the title of the manuscript to reflect more accurately the balance between development and applications. The title now reads: “Genetically encoded biosensor for branched-chain amino acid metabolism in yeast and applications for isobutanol and isopentanol production”

Right now, I am evaluating the manuscript according to its main aim as stated in the title, i.e. the development of a sensor. I am not sure whether I precisely understood how the sensor really

functions. The transcription factor binds to a metabolic intermediate. Thus, primarily the sensor should be a sensor for the concentration of this intermediate. However, throughout the text the authors also imply that the sensor reports on the flux through the pathway. On the practical level the sensor output also seems to correlate with the productivity (i.e. a rate) of the strains, but I cannot get my head around this. I feel something fundamental is missing here (or I might not have gotten it). Also, gene/protein expression and (thus also the GFP expression driven by the sensor) should be a function of the cellular growth rate. How does this factor in here? Would cells with different growth rate but otherwise same α -IPM concentration have the same sensor output? Also, I am a bit concerned about the fact that the sensor output (fluorescence) was measured during exponential phase (13 h), while the product titers were measured only after 48 h (which is surely not exponential anymore): What if all product was only produced in the late exponential/early stationary phase? Thus, I am wondering how does fluorescence in the exponential phase mechanistically correlate with the final product titers? Is the product synthesis rate constant in the whole cultivation period, i.e. until 48h? While the practical application cases shown by the authors show that the “sensor” indeed allows to pick up mutants with increased product titers, I feel that for publication in NComm, we need more mechanistic understanding on how the sensor works, and what it is actually reporting (an intermediate concentration, a flux) and why.

The reviewer raises interesting questions, which we are happy to address. Our biosensor is based on ample evidence from previous studies that the Leu3p transcriptional regulator is activated by α -IPM (Sze JY, et al. (1992) In vitro transcriptional activation by a metabolic intermediate: activation by Leu3 depends on alpha-isopropylmalate. *Science* 258(5085):1143-5; Zhou, K. M., Bai, Y. L., & Kohlhaw, G. B. (1990). Yeast regulatory protein LEU3: a structure-function analysis. *Nucleic Acids Res*, 18(2), 291-298. doi:10.1093/nar/18.2.291; Remboutsika, E., & Kohlhaw, G. B. (1994). Molecular architecture of a Leu3p-DNA complex in solution: a biochemical approach. *Mol Cell Biol*, 14(8), 5547-5557. doi:10.1128/mcb.14.8.5547; Wang, D., Zheng, F., Holmberg, S., & Kohlhaw, G. B. (1999). Yeast Transcriptional Regulator Leu3p. *Journal of Biological Chemistry*, 274(27), 19017-19024. doi:10.1074/jbc.274.27.19017; Wang, D., Hu, Y., Zheng, F., Zhou, K., & Kohlhaw, G. B. (1997). Evidence that intramolecular interactions are involved in masking the activation domain of transcriptional activator Leu3p. *J Biol Chem*, 272(31), 19383-19392. doi:10.1074/jbc.272.31.19383). Nevertheless, **we provide new measurements of intracellular concentrations of α -IPM, as the reviewer requested, and their correlation with biosensor output and BCHA production (new Supplementary Figure 4)**. We agree that gene/protein expression, including of GFP, should be a function of cellular growth rate, which we believe is part of the reason why measurements taken during the exponential phase of fermentation are most reliable; at least in part, because cellular growth rates are most homogeneous (in isogenic populations) and reproducible (across experiments) during this phase. It is difficult to answer whether cells with different growth rates but otherwise same α -IPM concentration would have the same sensor output, as it would depend on how much the activities of Leu3p and *LEU1* promoter (P_{LEU1}) vary with growth rate, which, to our knowledge, has not been reported, and fall outside the scope of this study. However, the high correlations of BCHA production to biosensor readout (Figure 1b,d), and low rates of false-negatives obtained in our demonstrations (**new Supplementary Figure 15**), amongst genetically diverse strains with different growth rates, suggest that this variation, if it exists, is probably not large enough to limit the utility of our biosensors.

It is also worth noting from our new intracellular α -IPM measurements, that there is a significant difference in α -IPM concentrations between an isobutanol-producing strain and a negative control, starting from the exponential phase (12h) to the stationary phase (30h), (**new Supplementary Figure 4a,b**). However, even though the α -IPM difference is larger in the stationary phase, the difference in biosensor output is larger during the exponential phase, which is consistent with our initial observation that data from the exponential phase is more reliable and reproducible. The rapid drop in biosensor output during the stationary phase is likely due to increased GFP degradation rate, exacerbated by the PEST tag, although other factors that may contribute by repressing Leu3p or P_{LEU1} activity in the stationary phase cannot be ruled out. These observations contrast with isopentanol-producing strains and the isopentanol-configured biosensor, in which the difference in α -IPM concentration between an isopentanol-producing strain and a negative control is not significant until the stationary phase measurement, yet the difference in biosensor output during the stationary phase is substantially larger than what is observed in the isobutanol-producing strain with the isobutanol-configured biosensor (**new Supplementary Figure 4d,e**). This is consistent with the fact that *LEU2* is deleted in the isobutanol-producing strains but not in the isopentanol-producing strains, which would be expected to cause α -IPM accumulation in the former but not the later strains. It is also consistent with the fact that the GFP reporter in the isopentanol-configured biosensor does not have a PEST tag, which makes its response more sensitive to low α -IPM concentrations and longer-lasting into the stationary phase. Interestingly, the isopentanol-configured biosensor is more sensitive to small differences in α -IPM concentrations in the lag and exponential phases of fermentation than our method using U-HPLC-orbitrap MS to measure intracellular α -IPM concentrations from cell cultures. The large difference in intracellular concentrations of α -IPM between the isobutanol and isopentanol strains, due to the different *LEU2* backgrounds, explains why the isobutanol-configured biosensor functions optimally with a PEST tag fused to the GFP reporter (to avoid high background and possibly early GFP saturation due to the higher α -IPM concentration), while the isopentanol-configured biosensor demonstrates increased sensitivity to lower α -IPM concentrations without a PEST tag.

The new data also explains why biosensor measurements taken during the exponential phase are most predictive of BCHA production even though those differences are better observed in measurements taken during the stationary phase (**new Supplementary Figure 4c,f**). During the exponential phase, both biosensor configurations display the highest sensitivity to small differences in α -IPM concentrations (as discussed above and shown in **new Supplementary Figure 4a,b,d,e**). Conversely, while BCHA production increases dramatically by the time the fermentation reaches stationary phase, both biosensor configurations are less sensitive to even large variations in α -IPM concentrations between strains at this late stage of the fermentation (possibly due, at least in part, to increased GFP degradation and extraneous Leu3p or P_{LEU1} regulation). BCHA concentrations at early to mid-exponential phase of fermentation (when biosensor readout is most predictive), are nonetheless lower (and thus more difficult to measure) than in stationary phase, likely due, at least in part, to lower cell concentrations and shorter time given to convert glucose to products in these early phases. This is especially true when comparing low cell density fermentations in 2% glucose (used in FACS to isolate high-producing strains and in the experiments shown in **new Supplementary Figure 4**) to high cell density fermentations in higher glucose concentrations (used in BCHAs production fermentations in most of the study). Nevertheless, biosensor outputs measured during the exponential phase can clearly predict

differences in the accumulated BCHA production after 24-48 hours of fermentation, including in high cell density fermentations. Therefore, we used biosensor measurements in the exponential phase of low cell density fermentations to isolate higher-producing strains, whose enhanced productivity we then confirmed in high cell density 48h fermentations, notably obtaining very low rates of false-positives (**new Supplementary Figure 15**).

In addition to the new data presented in the **new Supplementary Figure 4**, we have summarized these new mechanistic insights as a **new Supplementary Note 3** due to space constraints in the main text.

Finally, to avoid confusion, we have edited the manuscript, wherever appropriate, to say that the biosensors measure α -IPM concentrations or metabolic activity (Lines 75, 530), instead of metabolic flux.

Reviewers' Comments:

Reviewer #1:

Remarks to the Author:

All my previous concerns have been fully and satisfactorily addressed by the authors

Reviewer #2:

Remarks to the Author:

This is the revised version of a manuscript, which I reviewed more than a year ago. When I first read it, I was not very happy about several "bold" statements/conclusions and the absence of crucial experiments, which would support the overall message of the manuscript.

I am happy to see a fundamental improvement of the manuscript and new data making this paper much stronger. In particular the "Supplementary Notes" added to the supplementary information provide some useful insights to reader.

I think this manuscript can be accepted for publication in Nature Communications.

Here are my comments to the response to my initial major remarks where necessary:

1. Naming of biosensors.

The authors argue that "Biosensors are commonly named after the final product for which they are designed, even if their molecular mechanism is based on sensing a precursor metabolite." and present some examples (big names in big journals) where this was the case. Unfortunately, also Keasling and Church do not do this (anymore) (Keasling: e.g.

<https://pubs.acs.org/doi/10.1021/acssynbio.9b00292>; it would have been highly irritating if Church would have done this here: <https://www.ncbi.nlm.nih.gov/pmc/articles/PMC6079105/>) and other major hot spots of biosensor development and engineering (Denmark (btw. also Keasling's lab); Germany, South Korea) have never done so.

This makes totally sense. Just imagine: Somebody requests any of the two Leu3p-biosensors and would use biosensor in the context of a different final product. This scientist would (of course) have to rename it again when writing up the manuscript. How confusing would that be for all readers?

However, referring to different "configurations" does the trick.

2. False positives.

Thank you for the clarification, I was not precise. Indeed, I meant randomly mutated genomes. In the cases of randomly mutagenized genes, I would not have expected so many false-positive variants anyhow. The data presented in Suppl. fig. 15 is very convincing.

4. Sensor specificity.

Thank you for the huge amount of extra work. As expected, cellular uptake must have huge impact on the data obtained. Nonetheless, good that these experiments have been performed and included in the manuscript.

5. PEST-Tag

Thank you for expanding the text and performing additional experiments. It is good to see that addition of this tag apparently does not impede biosensor functionality.

7. Library size and screening

Of course – I was only interested in the original library size and the number of different variants finally obtained. The genetic complexity does only decrease during the enrichment steps anyway. I think that this additional data strongly improves the manuscript.

Response to reviewers' comments

We thank the reviewers for carefully reading our revised manuscript and the responses. Below we provide point-by-point responses to the reviewers' most recent comments, and descriptions of the changes we have made to the manuscript to address their remaining concerns.

Please note that changes to the manuscript are shown in the tracking record in the revised manuscript.

REVIEWER COMMENTS

Reviewer #1 (Remarks to the Author):

All my previous concerns have been fully and satisfactorily addressed by the authors

We thank the reviewer for the support.

Reviewer #2 (Remarks to the Author):

This is the revised version of a manuscript, which I reviewed more than a year ago. When I first read it, I was not very happy about several “bold” statements/conclusions and the absence of crucial experiments, which would support the overall message of the manuscript.

I am happy to see a fundamental improvement of the manuscript and new data making this paper much stronger. In particular the “Supplementary Notes” added to the supplementary information provide some useful insights to reader.

I think this manuscript can be accepted for publication in Nature Communications.

Here are my comments to the response to my initial major remarks where necessary:

1. Naming of biosensors.

The authors argue that “Biosensors are commonly named after the final product for which they are designed, even if their molecular mechanism is based on sensing a precursor metabolite.” and present some examples (big names in big journals) where this was the case. Unfortunately, also Keasling and Church do not do this (anymore) (Keasling: e.g. <https://pubs.acs.org/doi/10.1021/acssynbio.9b00292>; it would have been highly irritating if Church would have done this here: <https://www.ncbi.nlm.nih.gov/pmc/articles/PMC6079105/>) and other major hot spots of biosensor development and engineering (Denmark (btw. also Keasling's lab); Germany, South Korea) have never done so.

This makes totally sense. Just imagine: Somebody requests any of the two Leu3p-biosensors and would use biosensor in the context of a different final product. This scientist would (of course) have to rename it again when writing up the manuscript. How confusing would that be for all readers?

However, referring to different “configurations” does the trick.

We agree with the reviewer's comments and thank them for their support.

2. False positives.

Thank you for the clarification, I was not precise. Indeed, I meant randomly mutated genomes. In the cases of randomly mutagenized genes, I would not have expected so many false-positive variants anyhow. The data presented in Suppl. fig. 15 is very convincing.

We thank the reviewer for the support.

4. Sensor specificity.

Thank you for the huge amount of extra work. As expected, cellular uptake must have huge impact on the data obtained. Nonetheless, good that these experiments have been performed and included in the manuscript.

We thank the reviewer for the support.

5. PEST-Tag

Thank you for expanding the text and performing additional experiments. It is good to see that addition of this tag apparently does not impede biosensor functionality.

We thank the reviewer for the support.

7. Library size and screening

Of course – I was only interested in the original library size and the number of different variants finally obtained. The genetic complexity does only decrease during the enrichment steps anyway. I think that this additional data strongly improves the manuscript.

We thank the reviewer for the support.

At this time, Reviewer #3 was unable to comment so we asked an additional reviewer to comment: I think this new version is greatly improved, but also more complex now. The new title is indeed better, but way too long (not complying with Nature's guidelines for short titles) and has been shortened.

We have shortened the title to “Biosensor for yeast branched-chain amino acid metabolism and applications in isobutanol and isopentanol production”, which is now within the journal's word limit.

Thank you for the detailed answer regarding the mechanistic understanding of the role of Leu3p and the usefulness of fluorescence/products determination during the different stages of yeast growth. In principle, I think that the experiments performed to understand the correlations between sensor response and intracellular metabolite concentration are useful. However, the number of data points over the course of time is rather low for clear conclusions.

Taken together, the comments and questions have been properly addressed. I think that the manuscript can be accepted.

We thank the reviewer for the support. We agree that in general it is desirable to get as many data points as possible. However, in these new experiments we focused on gathering the highest quality data possible by using the same cell culture time courses (in triplicate) to make simultaneous measurements of biosensor fluorescence, isobutanol or isopentanol production, and α -IPM intracellular concentrations. This required us to use relatively large fermentation flasks (500 mL) in order to obtain enough cells for the metabolomic measurements, especially considering how dilute the cell cultures are in the initial time points. This limited the number of cultures that could be grown simultaneously and thus the number of samples and ultimately data points that we could collect. Nevertheless, the number of data points we obtained, while relatively few, are of sufficient quality to identify correlations and trends in α -IPM concentrations, biosensor readout, and isobutanol or isopropanol production, which support the proposed biosensor mechanism.